# The impact of GNSS Zenith Total Delay data assimilation on the short-term precipitable water vapor and precipitation forecast over Italy using the WRF model

Rosa Claudia Torcasio[1], Alessandra Mascitelli[1], Eugenio Realini[2], Stefano Barindelli[2], Giulio Tagliaferro[3], Silvia Puca[4], Stefano Dietrich[1], Stefano Federico[1]

[1] National Research Council of Italy, Institute of Atmospheric Sciences and Climate (CNR-ISAC), via del Fosso del Cavaliere 100, 00133 Rome, Italy
[2] Geomatics Research & Development srl (GReD), via Cavour 2, 22074 Lomazzo, Italy
[3] BIPM Time Department, Sèvre, France
[4] Civil Protection Department, via Vitorchiano 4, 00189 Rome, Italy

*Correspondence to*: Stefano Federico (s.federico@isac.cnr.it)

**Abstract.** The impact of assimilating GNSS-ZTD (Global Navigation Satellite Systems - Zenith Total Delay) on the precipitable water vapor and precipitation forecast over Italy is studied for the month of October 2019, characterized by several moderate to intense precipitation events, especially over north-western Italy. The WRF (Weather Research and Forecasting) model, version 4.1.3, is used with its 3DVar data assimilation system to assimilate ZTD observations from 388 GNSS receivers distributed over the country. The dataset was built collecting data from all the major national and regional GNSS permanent
networks, achieving dense coverage over the whole area. The water vapor forecast is verified for the forecast hours 1-6h after the last data assimilation time. Results show that WRF underestimates the atmospheric water vapor content for the period, and GNSS-ZTD data assimilation improves this underestimation. The precipitation forecast is verified in the phases 0-3h and 3-6h after the last data assimilation time using more than 3000 rain gauges spread over Italy. The application of GNSS-ZTD data assimilation to a case study improved the precipitation forecast by increasing the rainfall maximum and by better focusing the
precipitation pattern over north-eastern Italy, the main drawback being the prediction of false alarms. Considering the study over the whole period, GNSS-ZTD data assimilation had a positive impact on rainfall forecast, with an improvement of the performance up to 6 hours, and with statistically significant results for moderate to intense rainfall thresholds (25-30 mm/3h).

## 1 Introduction

The Mediterranean area is often struck by severe weather and deep convective events because of the presence of the warm sea, the complex orography of the area, and the specific synoptic scale environment. This scenario is worsened by the climate

change, which is affecting many weather and climate extremes, and the frequency and intensity of heavy precipitation events have increased in most of the world (Masson-DelMotte et al., 2021). Numerical Weather Prediction models (NWP) are useful tools to predict adverse weather conditions and to guide responsive actions for mitigating the impact of severe weather. Over the past years, the use of NWP models, along with an increasing availability of computing power, led to an improvement of the forecast accuracy. However, NWPs have well-known difficulties in representing the physical processes at small spatial and temporal scales, which are involved in convective or severe weather events (Stensrud et al., 2009).

One of the common problems of NWP-based nowcasting and short-term forecasting is the spin up time, because the model needs a few hours to balance the inconsistencies between the initial and boundary conditions to properly reproduce the small-scale dynamic (Lagasio et al., 2019). Data assimilation of local observations in NWP has been reported as a key factor to reduce this issue and to improve the prediction of high impact weather events (Federico et al., 2021). Among local observations, water vapor plays a key role for its importance in humid and energetic exchanges in the atmosphere. Therefore, a good knowledge of water vapor distribution in space and time is a fundamental requirement for improving NWP forecasts of convective and severe weather events.

Global Navigation Satellite System (GNSS – collective term used to address all global and regional satellite navigation systems, including GPS, Galileo, GLONASS and BeiDou) routine observations processing for geodetic and geophysical purposes can provide estimates of the tropospheric delays (generally ZTD-Zenith Total Delay), directly connected to the water vapor content in the atmosphere, which can be very useful to improve the NWP forecast. Relevant, albeit non exhaustive, experiences are shortly summarized hereafter:

- Vedel and Huang (2004) assimilated GNSS-ZTD into the HIgh Resolution Limited Area Model using three dimensional variational data assimilation (3DVar) and found improvements for the forecast of geopotential height and high precipitations.
- Poli et al. (2007), assimilated the GNSS-ZTD using four dimensional variational data assimilation (4DVar) and the ARPEGE (Action de Researche Petite Echelle Grande Echelle) global model. Results show the positive impact of the GNSS-ZTD data assimilation on the forecast of synoptic-scale circulations and precipitation in spring and summer. Other studies followed in France (Boniface et al., 2009; Yan et al., 2009) and found a positive impact of GNSS-ZTD data assimilation on the NWP forecast.
- Bennitt and Jupp (2012) assimilated GNSS-ZTD observations by both 3DVar and 4DVar using the Met Office North Atlantic and European model. The assimilation of GNSS-ZTD resulted in an improvement of the cloud forecast.
- Lindskog et al (2017) performed GNSS-ZTD data assimilation into the HARMONIE-AROME model at 2.5 km horizontal resolution. The assimilation was performed by 3DVar and improved the forecast up to one and a half day.
- Rohm et al. (2019) assimilated GNSS (both ZTD and Precipitable Water) in the WRF model at 4 km horizontal resolution over Poland for two months. They found an improvement in predicting both water vapor and precipitation. In the same direction, Trzcina et al. (2020) showed the positive impact of assimilating the GNSS tomography wet refractivity field on temperature and precipitation forecast at the short-range (0-6h) over Poland.

- Giannaros et al. (2020) showed the positive impact of GNSS-ZTD data assimilation on both precipitation and water vapor forecast over Greece, and Caldas-Alvarez et al. (2020), showed similar results with the climatic setting of the COSMO model over a large portion of the Mediterranean and Central Europe.

- Risanto et al. (2021) assessed the impact of Global Positioning System (GPS) precipitable water vapor (PWV) data assimilation on short-range North American monsoon precipitation forecasts over Nortwest Mexico. They showed that GPS-PWV data assimilation created more favorable conditions for nocturnal convection organization leading to a better precipitation forecast.Singh et al. (2019) studied the impact of GNSS-ZTD data assimilation over the Indian Region with the WRF model. They shows a positive impact on the precipitation forecast and on the lower to middle tropospheric moisture, upper air temperature, and middle and upper tropospheric wind.

- Yang et al. (2020) assimilated both radar reflectivity and GNSS-ZTD for a heavy rainfall event over Taiwan. They showed the complementary role of both observations in improving the precipitation forecast.

The assimilation of GNSS data is also used in the operational context. The Rapid Refresh model over US assimilates GPS-derived Integrated Water Vapor (IWV) every hour from 300 stations across the US (Benjamin et al. 2016). The study shows that there is a clear benefit in using GNSS observations in rapid refresh weather forecast.

Considering the GNSS data assimilation over Italy, Faccani et al. (2005), using MM5 model at 9 km horizontal resolution and 3DVar to assimilate GNSS-ZTD over Italy, found improvements in the precipitation forecast during the transition from winter to spring.

By assimilating a wide range of Sentinel-1 and GNSS-ZTD observations over Italy in the WRF model, Lagasio et al. (2019) found that the forecasts benefit the most when the model is provided with information on the wind field and/or the water vapor

content. Mascitelli et al. (2019, 2021) reported two successful experiments of GNSS-ZTD and PWV data assimilation with the RAMS@ISAC model in Italy. The 3DVar was used to assimilate GNSS-ZTD, while nudging was used to assimilate the PWV. In both cases the assimilation showed a significant improvement in the short-term prediction of water vapor with smaller impact on the precipitation forecast.

This paper goes in a similar direction in the sense that it uses the GNSS-ZTD data assimilation to improve the precipitation

and water vapor forecast over Italy. It uses a period of one month (October 2019) and the data of 388 GNSS receivers widespread over the country, giving a robust assessment on the impact that GNSS-ZTD data assimilation can have on the forecast at the local scale. In addition, for the first time over Italy, the sensitivity of the results to the number of GNSS receivers used (data thinning) and to the bias correction are shown for a subperiod of 16 days (5-10 and 14-23 October).

This paper is organized as follows: Section 2 shows the data and methods used including the model setting, the GNSS-ZTD

dataset, and the statistics for the verification. Section 3 discusses the results showing the impact of GNSS-ZTD data assimilation on the WRF analyses (both ZTD and PWV), on the precipitation forecast for a case study, and on the PWV and precipitation forecast for the whole period. Conclusions are given is Section 4. Finally, more details on the statistics used in the paper as well as the resampling test are reported in two appendices.

This paper has a supplement were we consider: a) the precipitation for October 2019 and compare this precipitation with the average for the period 2011-2022; b) the distribution of the rainfall intensities for the period October 2019 and the correspondent distribution for the period 2011-2022; c) a short synoptic-scale analysis of October 2019; d) the rainfall prediction scores for a case study.

## 2 Data and methods

### 2.1 WRF model configuration and assimilation method

The model used in this study is the Weather Research and Forecasting with advanced WRF dynamic (WRF-ARW, Version 4.1.3, Skamarock et al., 2019). The model was run over one domain covering the whole Italian territory and the Central Europe (Figure 1) with a spatial horizontal resolution of 3km. The model grid has 635x635 grid points and 50 vertical levels, with the model top at 50 hPa. The main physical parameterization used are the following: the Thompson scheme (Thompson et al. 2008) for microphysics, the Mellor-Yamada-Janjic (Eta) TKE scheme for boundary layer (Janjic, 1994) the Dudhia scheme (Dudhia, 1989) and the Rapid Radiative Transfer Model (RRTM, Mlawer et al. 1997) for short wave and longwave radiation, respectively.

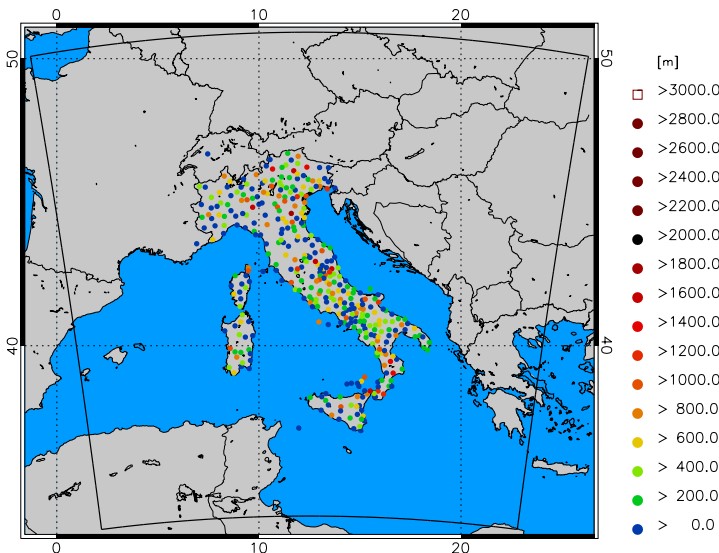

**Figure 1: WRF model domain and GNSS receivers height ([m]) above sea level.**

The experiment aims at evaluating the impact of GNSS-ZTD data assimilation on the precipitation prediction at the short-term (up to 6h in this paper). For this purpose, we considered a one-month period, from 02 to 31 October 2019. The choice of the period is due to two main reasons: on the one hand, this month was characterized by both widespread and localized precipitation

events over Italy, on the other hand, a dense network of GNSS receivers (about 500) is available for the country. It is important to note that the number of receivers actually used for data assimilation was reduced to 388 by applying data thinning as discussed in Sect. 2.2.

GNSS-ZTD data assimilation was performed using the 3DVar tool distributed with WRF model, which is one of the components of WRFDA system, that includes also 4DVar and Ensemble data assimilation systems (Barker et al., 2004; Barker et al. 2012; Huang et al., 2009).

We consider two kinds of simulations: control simulations, without GNSS-ZTD data assimilation, hereafter also CTRL, and simulations assimilating GNSS-ZTD, hereafter also GNSS. The European Centre for Medium range Weather Forecast (ECMWF) Integrated Forecast System (IFS) 3-hourly operational analysis/forecast cycle at 0.25° starting at 12 UTC on the day before the actual day to forecast is used for initial and boundary conditions, to simulate a real forecasting context. The temporal scheme used for the simulations uses a Very Short-term Forecast (VSF) approach, with a 6h update (Figure 2).

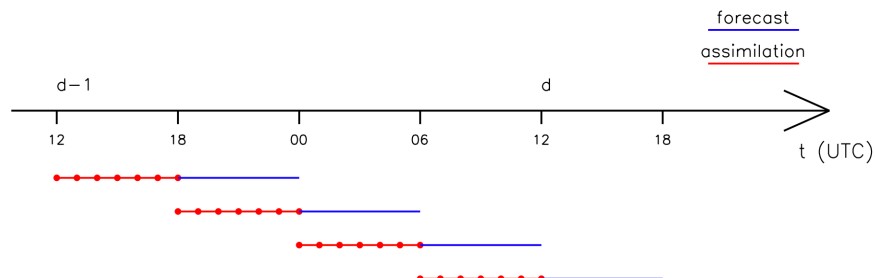

Figure 2: Rapid Update Cycle at 6h. Red dots denote analysis times.

In this scheme, for each day, we run four simulations starting from a cold start. Each simulation lasts 12 hours. The first 6h of each run are used for the model spin-up and for data assimilation in GNSS simulations, while the last 6h are used as forecast. Therefore, 4 runs are necessary to cover a whole day. For GNSS simulations, in the assimilation phase, we considered an analysis every 1 hour (red points of Figure 2), starting from the beginning of the simulation and reaching the 6th hour, so a total of 7 analyses are performed for each run. For CTRL simulations we use only initial and boundary conditions from ECMWF-IFS and no other data are assimilated.

GNSS-ZTD observations are assimilated by 3DVar, which is a variational approach involving the calculation of the analysis that minimizes a cost function by measuring its distances from background and observations. The cost function is given by:

$$J(x) = \frac{1}{2}(\boldsymbol{x} - \boldsymbol{x_b})^T \boldsymbol{B}^{-1}(\boldsymbol{x} - \boldsymbol{x_b}) + [\boldsymbol{y_o} - H(\boldsymbol{x})]^T \boldsymbol{R}^{-1}[\boldsymbol{y_o} - H(\boldsymbol{x})]$$

where $\boldsymbol{x}$ is the state vector, $\boldsymbol{x_b}$ is the background field, $H$ is the forward observational operator which transforms the $\boldsymbol{x}$ vector in the observational space, $\boldsymbol{y_o}$ is the observations vector, $\boldsymbol{B}$ and $\boldsymbol{R}$ are the error covariance matrices, respectively for background and observations.

The **R** matrix is given by the sum of instrumental and representation errors. To prepare observational data for 3DVar, the OBSPROC tool was employed for GNSS-ZTD observations. Data for each analysis were considered on a 1-hour time range, centered at the hour of the analysis. The **R** matrix is diagonal, which requires data thinning (see next section), and the ZTD error is set to 5 mm. Different regional networks are considered to reach the considerable number of GNSS-ZTD receivers used in this paper; however, we used a constant value for the errors of all receivers because the software and the processing method are the same for all the receivers. Also, the GNSS-ZTD time series were visually checked and no specific differences among networks arose. The value of 5 mm was not specifically computed for this experiment but comes from previous comparisons that, in any case, do not extend to the whole Italy (Tagliaferro, 2021; Krietemeyer et al. 2018.; Mascitelli et al. 2019 and 2021). In these works, the GNSS-ZTD retrieved with the method used in our paper was compared with other methods and with radiosoundings. In general, comparison with radiosondes shows differences in the range 1.0-1.5 cm (i.e. larger than the error used in this paper), while comparison with other methods shows differences between 0.1 and 0.8 mm. Now, the comparison with radiosondes is less representative of the GNSS-ZTD error because radiosondes can move far from the GNSS receiver, and the 0.5 cm used in this paper comes from the comparison with other methods to estimate ZTD. However, future experiments considering different errors for different networks should be done to assess more in detail this point.

As regards the **B** matrix, the calculation for this study is performed via the GEN_BE tool, employing the NMC method (Parrish and Derber, 1992) for the month of October and the option CV5, which accounts for five control variables.

The GNSS-ZTD observations are considered to have unbiased errors compared to the WRF model. To achieve this goal, a statistical bias correction was applied to the ZTD data for the whole period. First the raw GNSS data are assimilated in the 3DVAR to calculate the corrections that come from the background. The difference between the observation and the background is saved for each receiver and for each time giving the quantity $(O\text{-}B)_{k,t}$, where $k$ is the receiver index and $t$ is the time. The quantity $(O\text{-}B)_{k,t}$ takes into account for the difference between the model orography and receiver height that, in our case, is never larger than 300 m (see Section 2.2.1). For each receiver we compute the background bias by averaging $(O\text{-}B)_{k,t}$ over all times:

$$\overline{(O-B)_k} = \sum_{t=1}^{N} \frac{(O-B)_{k,t}}{N}$$

Where $N$ is the total number of times (i.e. observations) available for each GNSS receiver. Then we use the corrected observation $O'_{k,t}$ in the 3D-Var:

$$O'_{k,t} = O_{k,t} - \overline{(O-B)_k}$$

This method is like that applied in many papers including some cited in the Introduction.

## 2.2 Observational datasets and verification procedure

### 2.2.1 GNSS data

To achieve a good spatial density of the ZTD measurements, the GNSS data have been collected from all major Italian national and regional GNSS networks. To derive ZTDs, GPS L1 and L2 pseudorange and phase measurements were adjusted in PPP mode (Bevis et al., 1992; Zumberge et al., 1997), by using the GNSS data processing suite Breva, developed by GReD, based on the open-source software goGPS (Herrera et al., 2016).

Several of the used stations tracked only GPS satellites, thus we decided to process only GPS observations, disregarding data
from other GNSS constellations. Table 1 describes the corrections applied to the observations, as well as the stochastic model.

**Table 1: Corrections applied to the observations and stochastic model.**

| | |
|---|---|
| Ionospheric delay | Pre-eliminated through the iono-free linear combination of L1 and L2 |
| Ephemerides | IGS final combination |
| Coordinates | Estimated, one set per station per day |
| ZTD | Estimated epoch wise, with random walk model (noise 0.003 $m/\sqrt{h}$) |
| ZTD North and East gradient | Estimated epoch wise, with random walk model (noise 0.0001 $m/\sqrt{h}$) |
| Solid Earth | Corrected according to IERS 2010 |
| Ocean loading effects | Corrected using coefficient computed from FES 2004 model |
| Mapping function | Vienna Mapping Function |

Figure 1 presents the locations and the heights of the 388 GNSS stations used over the Italian peninsula and its surroundings
after a reduction of their number. Although the initial number of GNSS receivers was about 500, more than 100 GNSS receivers were discarded considering the following two requirements: a) the difference between the receiver and WRF model height at the closest grid point to the GNSS receiver must be less than 300 m (similarly to Bennitt and Jupp, 2012; Rohm et al., 2019; and Mascitelli et al., 2019) and b) in case two or more receivers fell in the same WRF grid cell, the one whose height was closer to the model orographic height was retained. At the end of the process, 388 GNSS receivers were used for data
assimilation and the closest distance between any two receivers was larger than 10 km. No further data thinning was applied.

### 2.2.2 Forecast verification

As regards the evaluation of the precipitation forecast performances, five precipitation scores are calculated, i.e. Frequency Bias (FBIAS), Probability of Detection (POD), Threat Score (TS), Equitable Threat Score (ETS) and False Alarm Rate (FAR). A detailed description of these scores is provided in Appendix A.

In the following, we will show the results of the scores summarizing them through the Performance Diagram (Roebber, 2009), with the exception of ETS. In this diagram the perfect score is the one that reaches the upper right corner of the diagram. The x-axis represents the Success Ratio (SR) which is defined as 1-FAR, while the POD is on the y-axis. The straight lines from the origin represent the FBIAS while the hyperboles branches are the TS. We considered the forecast for both types of run, CTRL and GNSS, taking into account two main periods after the last analysis time, in order to evaluate the impact of GNSS-ZTD data assimilation on the precipitation forecast at different ranges: the first 3h (from the 6th to the 9th hour of run) and the second 3h of forecast (from the 9th to the 12th of run). Model forecast in correspondence of a given rain gauge is computed using the nearest-neighbour method. By this method, we consider all WRF precipitation values at grid points in a radius of $2\Delta x\sqrt{2}$ from the rain gauge, $\Delta x$ being the WRF model grid spacing, and we select the grid point with closest rainfall value to the rain gauge observation.

Precipitation data come from the Italian rain gauge network, with more than 4000 rain gauges over Italy. This network belongs to the Italian regional administrations and data are collected nationwide by the Department of Civil Protection (Davolio et al., 2015).

Furthermore, a test to assess the statistical significance of the differences between CTRL and GNSS precipitation forecasts is performed. The resampling test of Hamill (1999) is used (see Appendix B for details).

For precipitable water vapor verification, we focused on both the assimilation and forecast phases. Verification is made calculating two main scores:

- BIAS, which measures the mean differences between forecast (F) and observation (O):

$$BIAS = \frac{1}{N}\sum_{i=1}^{N}(F_i - O_i) \tag{1}$$

- Root Mean Square Error (RMSE), which measures the mean of the squared differences between forecast and observation:

$$RMSE = \sqrt{\frac{1}{N}\sum_{i=1}^{N}(F_i - O_i)^2} \tag{2}$$

with $N$ number of forecast-observation pairs considered in the statistics.

## 3 Results and discussion

### 3.1 ZTD and precipitable water vapor results for the analysis phase

In this section we consider the differences between the first guess (FG) and the observations, and between the analyses (ANL) and the observations. Statistics are presented considering all simulations (30*4). For each simulation, six times are considered for first guess and analysis, one for each time (red dots of Figure 2), except for the analysis at the initial time of each simulation because it is coincident with the last analysis time of the previous simulation, the latter being considered in the results.

Two types of statistics are shown. In the first statistics, we show the time series of the BIAS and RMSE errors aggregating all receives together for each analysis time; in this case, referring to the Eqn. (1) and (2), N is the number of GNSS receivers considered at the analysis time. In the second statistics, we show the PWV BIAS and RMSE computed for each receiver; in this case, referring to Eqn. (1) and (2), N is the number of analyses done for the whole period (30*4*6=720 first guess and analysis pairs).

Figure 3 shows the time series for ZTD BIAS (Figure 3a) and ZTD RMSE (Figure 3b) calculated for the whole period. From Figure 3a it is apparent that the ANL BIAS (blue curve) has lower absolute values compared to that of FG (red curve) for all times considered. The better results obtained by ANL is confirmed also by the BIAS values calculated over all times together, shown in the upper right part of the figure, which is halved after the analysis (BIAS_A, -0.1 cm) compared to the first guess (BIAS_F, -0.2 cm). Interestingly, the WRF first guess bias is mainly negative showing an underestimation of the water vapor

content in the atmosphere. This underestimation is reduced by GNSS-ZTD data assimilation that increases the water vapor in the model. The positive impact of the GNSS-ZTD data assimilations is confirmed by the RMSE calculation (Figure 3b). The RMSE for the whole period for the analyses (RMSE_A) is 0.7 cm, compared to the value of 1.3 cm for the first guess (RMSE_F). The substantial impact of GNSS-ZTD data assimilation on the ZTD or water vapour simulation during the analysis phase is well known (Arriola et al., 2016; Poli et al., 2007; Mascitelli et al., 2019) and it is confirmed in this study.

a)

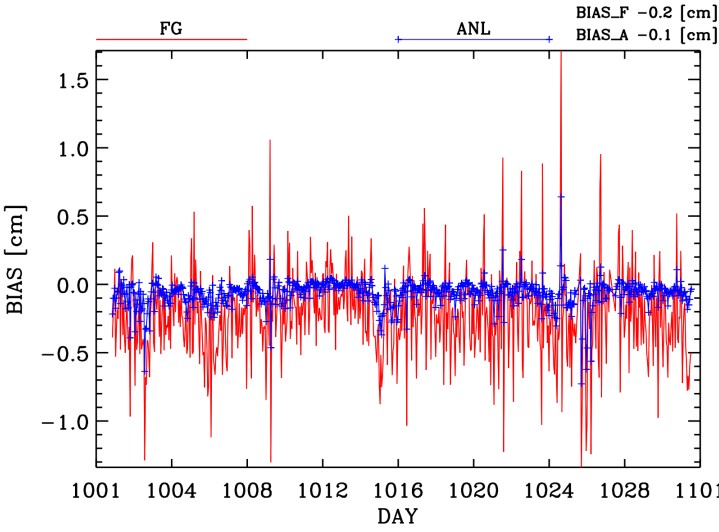

b)

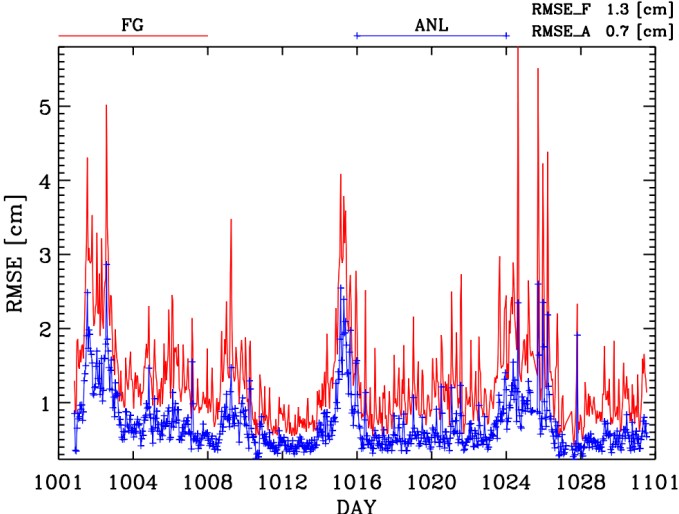

Figure 3: Time series of ZTD BIAS (a) and of ZTD RMSE (b) calculated for all sensors together for first guess (red curve) and
analysis (blue curve) during the analysis phase. Dates are shown along the x-axis.

It is important to discuss the impact of GNSS-ZTD data assimilation over the Italian territory. This is shown considering the
PWV estimated by the ZTD. The PWV in millimeters is given by:

$$PWV = Q(ZTD_{GNSS} - ZHD_{WRF})$$

where $ZHD_{WRF}$ is the hydrostatic delay calculated using the Saastamoinen (1972) equation given the WRF surface pressure
($p_{sfc}$), latitude ($\phi$) and height (h):

$$ZHD_{WRF} = \frac{0.0022767 p_{sfc}}{1.0 - 0.0266 \cos(2\phi) - 0.00000029\, h}$$

Similarly to other studies (for example, Rohm et al., 2019), we estimate the ZHD from the WRF surface pressure because
pressure observations aren't available in correspondence of the GNSS-ZTD receivers.

The proportionality factor Q is computed as follows:

$$Q = \frac{10^6}{R_W(\frac{k_3}{T_m} + k'_2)}$$

where $R_w$=461 J/(kgK) is the gas constant for water vapor, $k'_2$=22.9726 K/hPa and $k_3$=375463 K²/(hPa) are the refractivity
constants from Reuger (2002) and $T_m$ is the mean temperature given by:

$$T_m = 70.2 + 0.72 T_{WRF}$$

where $T_{WRF}$ is the WRF surface temperature.

Figure 4 shows the PWV RMSE (panels a and b, for first guess and analysis, respectively) and the BIAS (panels c and d for
first guess and analysis, respectively) for the whole period at each GNSS receiver. Both statistics are improved by data
assimilation. The impact of GNSS-ZTD data assimilation on PWV RMSE is apparent as most of the green-orange-red colors

(RMSE between 1.0 and 2.5 mm, Figure 4a) are reduced to dark green-green colors (RMSE between 0.5 and 1.5 cm, Figure

4b).

For the BIAS the improvement is less apparent, but it is clearly shown by the increase of the orange, red and yellow dots in panel d) compared to panel c) of Figure 4.

a)

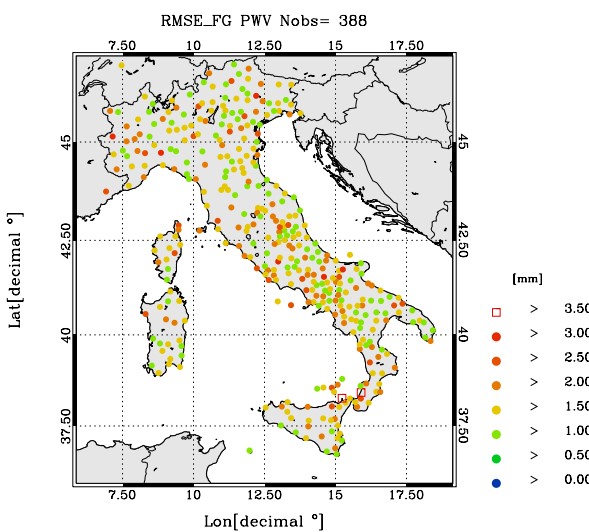

b)

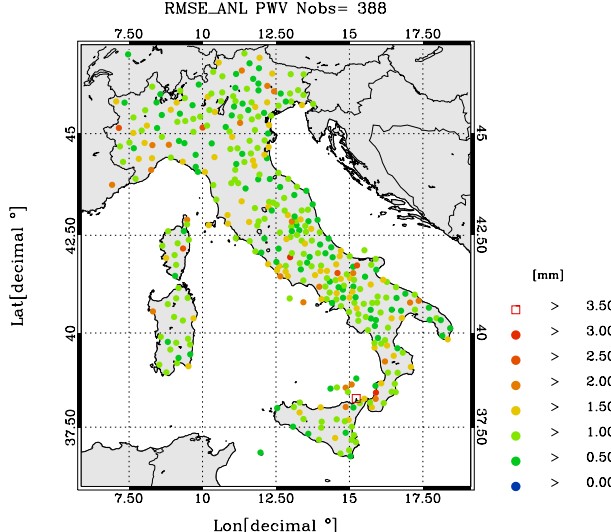

c)

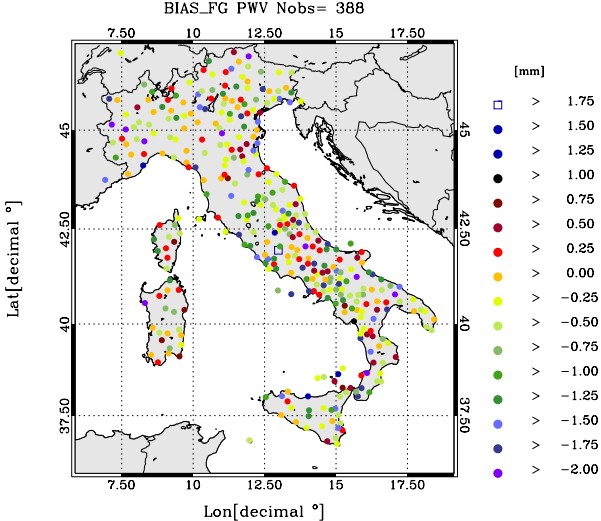

d)

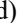

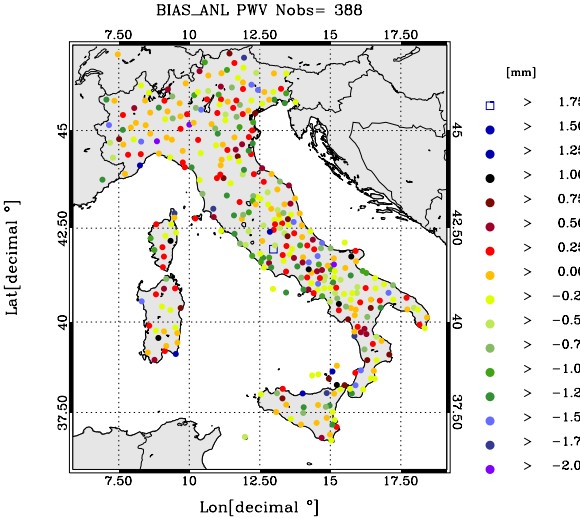

**Figure 4: PWV RMSE for FG (a) and ANL (b) and PWV BIAS for FG (c) and ANL (d) calculated for each sensor over all times of analysis.**

All in all, the results of this section show a positive impact of GNSS-ZTD data assimilation on the ZTD and PWV over the period, and the net result is the increase of the model water vapor content over the area, because the first guess underestimation of the ZTD (and PWV) is reduced by data assimilation.

## 3.2 Rainfall forecast for a case study

The selected date is the 15 October 2019, in the 3 hours between 18 and 21 UTC. This phase was selected because it is representative of the improvement that the assimilation of GNSS-ZTD had for several cases on the 3h precipitation forecast. During this time period, three main thunderstorms developed: the first two over the North-East and North-West of Italy, respectively, with a precipitation maximum between 60 and 70 mm/3h, and the third in Southern Italy, with precipitation ranging between 40 and 50 mm/3h. We start examining the innovations, i.e. the analysis minus first guess fields, at 18 UTC on 15 October (Figure 5a), which is the last analysis before the 3h forecast considered in this section and has an important role on the 3h rainfall forecast. Indeed, as shown below, the precipitation between 18 and 21 UTC has several correspondences with the innovations at 18 UTC.

Figure 5a shows the innovations at 18 UTC on 15 October 2019 at about 1800 m above the ground surface. A complex pattern of positive and negative innovations is shown thanks to the many observations available, which add information on the water vapor field at the subregional scale (the distance between two closest maxima or minima of Figure 5a can be roughly estimated in 50-70 km).

Figure 5b shows the latitude-height cross section of the innovations at the same time of Figure 5a. The cross section is for the longitude 12.6°E. Water vapour values in the range 0.4-1.0 g/kg are shown in several parts of the cross section at about 1600 m height a.g.l. in western Sicily (37.5 °N) revealing a considerable impact of GNSS-ZTD data assimilation (> 10% of the first guess value) on the water vapor field.

a)

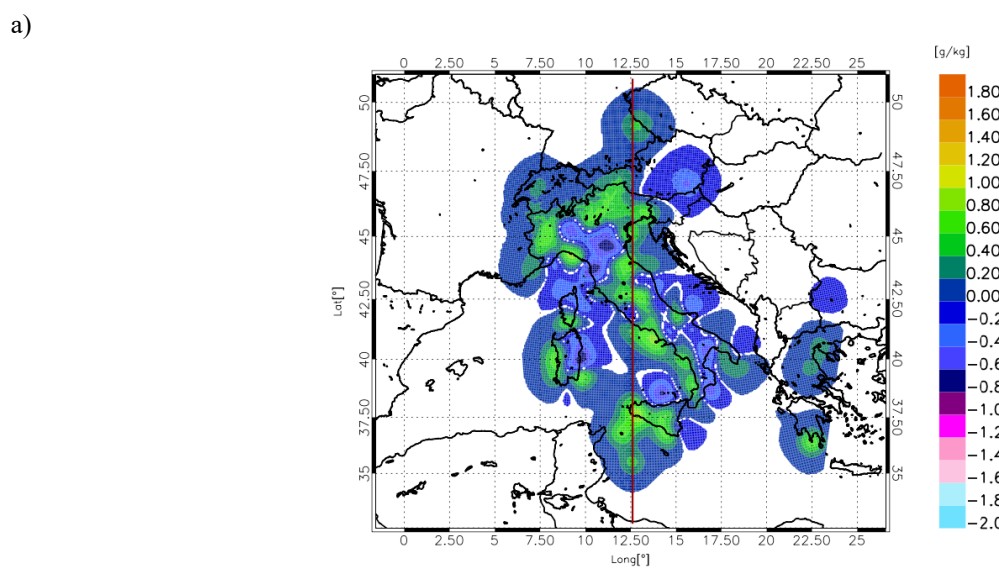

b)

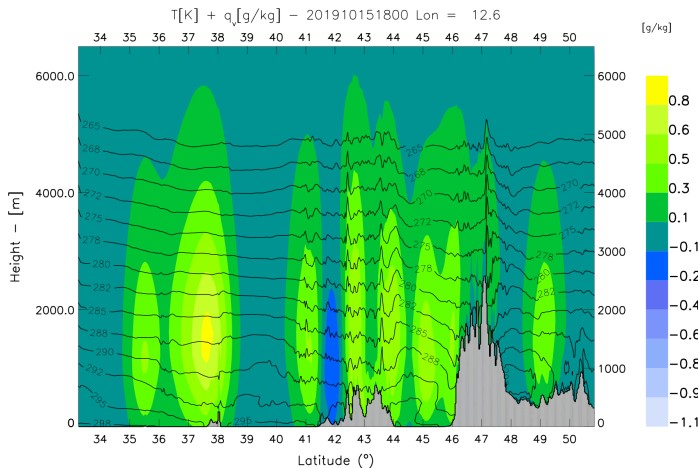

**Figure 5: Innovations of water vapor at 18 UTC on 15 October 2019; a) horizontal map around 1800 m above the surface. Cross section of panel b) is taken along the red line; b) latitude-height cross-section of water vapor innovations with the first guess temperature (black contours).**

Between 18 and 21 UTC, moderate to intense precipitation is shown in NE and NW of Italy by rain gauge observations, with maximum intensities larger than 60 mm/3h (Figure 6a). Some areas of moderate precipitation are apparent in Central and Southern Italy with maxima between 20 and 50 mm/3h.

    The CTRL, Figure 6b, has a good forecast because it represents the two main precipitation areas over NW and NE of Italy. There are, however, less satisfactory points in the CTRL rainfall forecast: first, the maximum over NE of Italy is composed by

two branches and one of them is displaced close to the sea, starting from Venice and going towards NE. This branch is a false alarm, at least in its southernmost part. Second, the maximum over Central Italy, starting from Tuscany and displaced in the SW-NE direction is displaced to the North, compared to observations. Third, the precipitation over Southern Italy is missed. The assimilation of GNSS-ZTD improves the precipitation forecast. The pattern of the precipitation over NE is more in agreement with observations because precipitation does not appear separated in two branches as in the CTRL and the

precipitation amount for GNSS simulations is higher, better catching the observed maximum. This agrees with the increase of the water vapor over NE of Italy given by the last assimilation (Figure 5a). A similar improvement is apparent in the South of Italy, where the precipitation was missed by CTRL. In this case the observed precipitation is well forecast by GNSS and the increase of precipitation is determined by the increase of water vapor over the area (Figure 5a). The maximum over Central Italy is not much improved by the assimilation of GNSS-ZTD because, even if there is better superposition with observations

compared to the control forecast, there are more false alarms in the GNSS forecast. The results of this case study are in line with those shown in Lagasio et al. (2019) for two severe precipitation events over Italy, showing the ability of GNSS-ZTD data assimilation to better focus the intense rainfall where observed.

a)

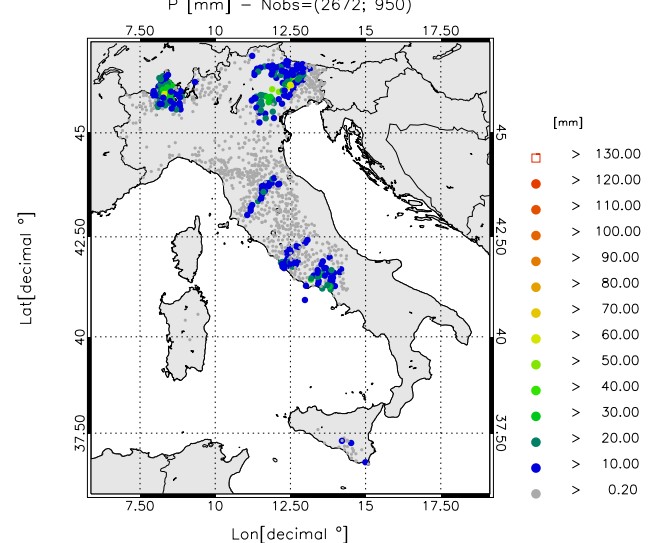

b)

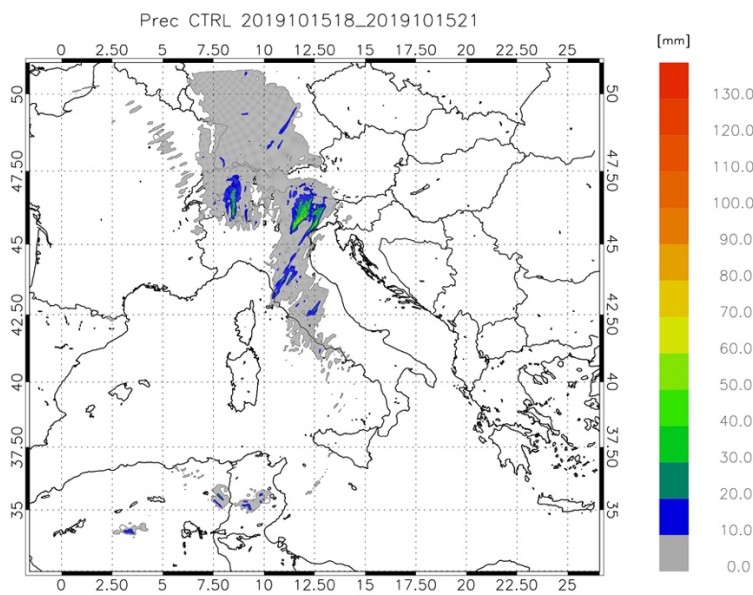

c)

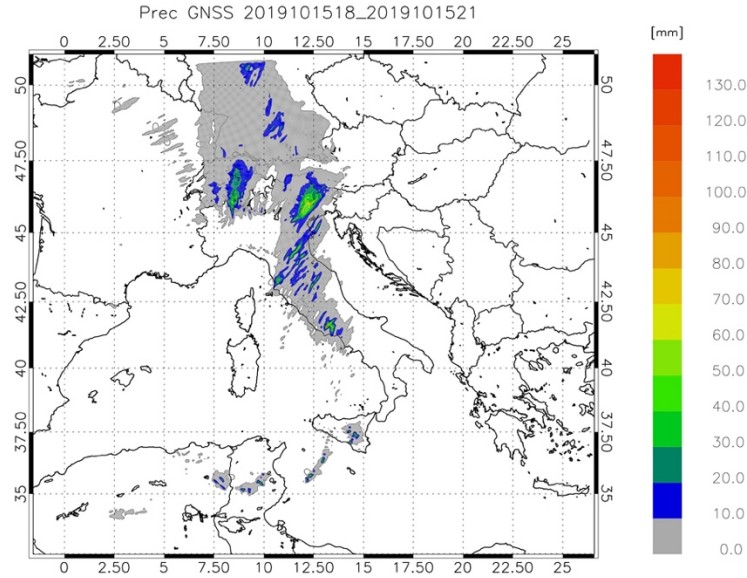

**Figure 6: Precipitation between 18 and 21 UTC on 15 October 2019: a) Rain gauges observations; b) CTRL forecast; c) GNSS forecast.**

Section S.3 of the supplemental material shows the rainfall prediction scores for CTRL and GNSS for this case study. GNSS forecast has a better performance compared to CTRL, especially for thresholds larger than 40 mm/3h.

### 3.3 Results for the water vapor forecast for the whole period

In this section we show the impact of GNSS-ZTD data assimilation on the PWV forecast. Three different statistics are considered: the time series of the differences between the PWV RMSE of CTRL and GNSS simulations for different forecasting times; the time series of the difference between the PWV absolute value of the BIAS of CTRL and GNSS simulations for different forecasting times; the RMSE maps of PWV for CTRL and GNSS on two different forecasting times. The PWV is computed as shown in Sect. 3.1, replacing $ZTD_{GNSS}$ with $ZTD_{WRF}$.

The first two statistics are shown in Figure 7 for the first (red curve), third (blue curve) and sixth hour of forecast (green curve). Considering the differences between CTRL and GNSS PWV RMSE, positive values show better performance of the forecasts assimilating GNSS-ZTD and, from Figure 7a, it is apparent that the GNSS-ZTD data assimilation improves the PWV forecast at different forecast ranges, with the exceptions of few cases. The RMSE of the GNSS simulations can have a RMSE lower than CTRL simulations up to 2.5 mm and, from Figure 7a, it is also notable a decrease of the improvement of GNSS-ZTD data assimilation for longer forecasting times as the red curve shows, with some exceptions, larger values than the blue curve, which, in turn, has larger values that the green curve. However, it is important to note that the positive impact of GNSS-ZTD data assimilation on the PWV forecast is still apparent after six hours of forecast.

The bias is also improved by the GNSS-ZTD data assimilation as the difference of the absolute value of the bias for CTRL and GNSS simulations shows positive values. As for RMSE, the bias improvement decreases with forecasting time. A similar

decrease of the improvement with forecasting time was shown by Yanghzao et al. (2023); conversely Rohm et al. (2019) showed an increase of the improvement after 9h leading time. Likely this behaviour depends on the period analysed, station density, domain extension and weather systems.

a)

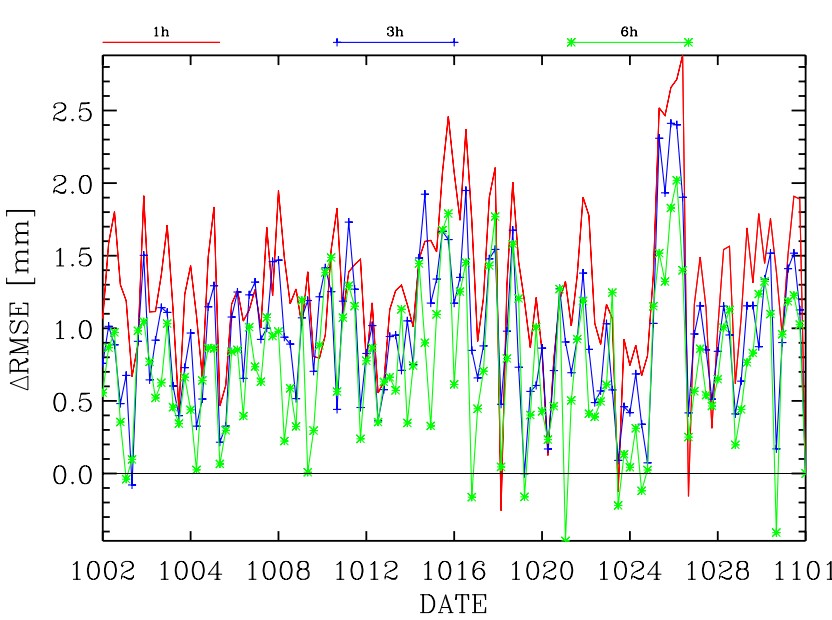

b)

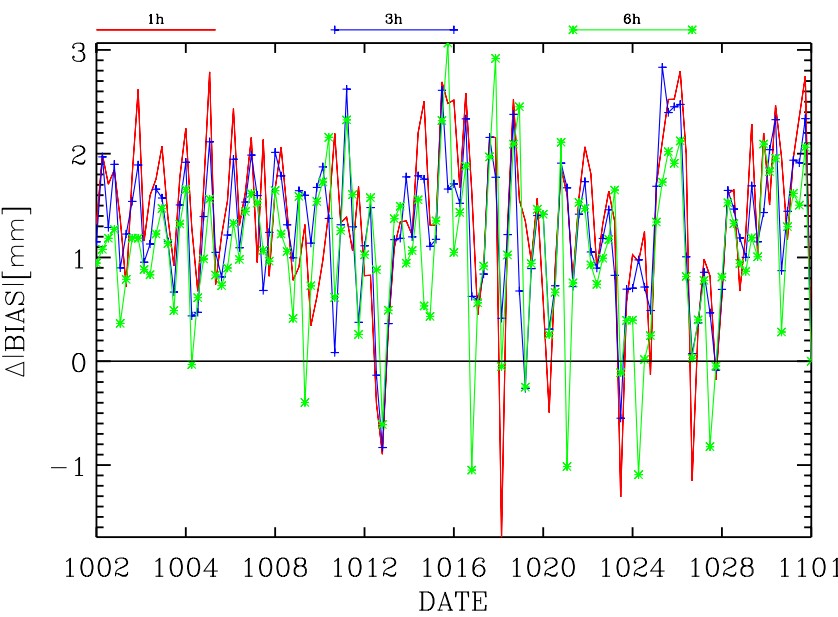

**Figure 7: Time series of the differences between PWV-RMSE for CTRL and GNSS (a) and of the difference of the absolute value of the Bias for the CTRL and GNSS simulations (b). Red curve is for the first forecast hour, blue curve is for the 3rd forecast hour and the green curve is for the 6th forecast hour.**

It is interesting to evaluate the spatial distribution of the improvement of GNSS-ZTD data assimilation on the PWV forecast over Italy. This is shown in Figure 8 for the first hour of forecast (panels a) and b) for CTRL and GNSS, respectively) and for the sixth hour of forecast (panels c) and d) for CTRL and GNSS, respectively). Figure 8 shows that the improvement of RMSE is not limited to a specific area, thanks to the good coverage of the GNSS receivers used in this work, but it is widespread with RMSE more than halved in correspondence of most receivers.

The RMSE for the sixth forecast hour increases for both CTRL and GNSS simulations compared to the first forecast hour, as expected because the forecast error increases with forecasting time. However, the increase of the RMSE error is larger for the GNSS simulations (comparison between panels b) and d)) than for the CTRL simulation (panels a) and c)), and the impact of GNSS-ZTD data assimilation decreases with the forecasting time over the whole Italian territory.

a)

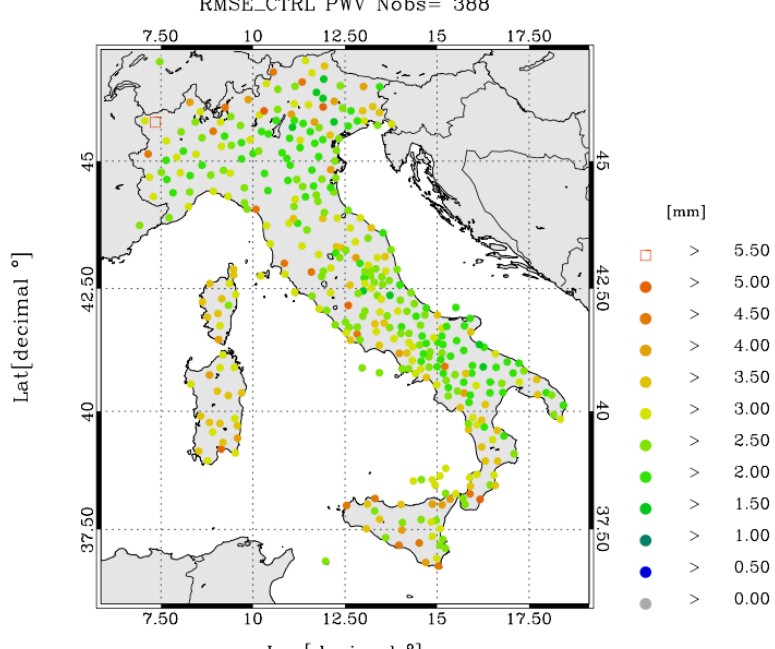

b)

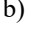

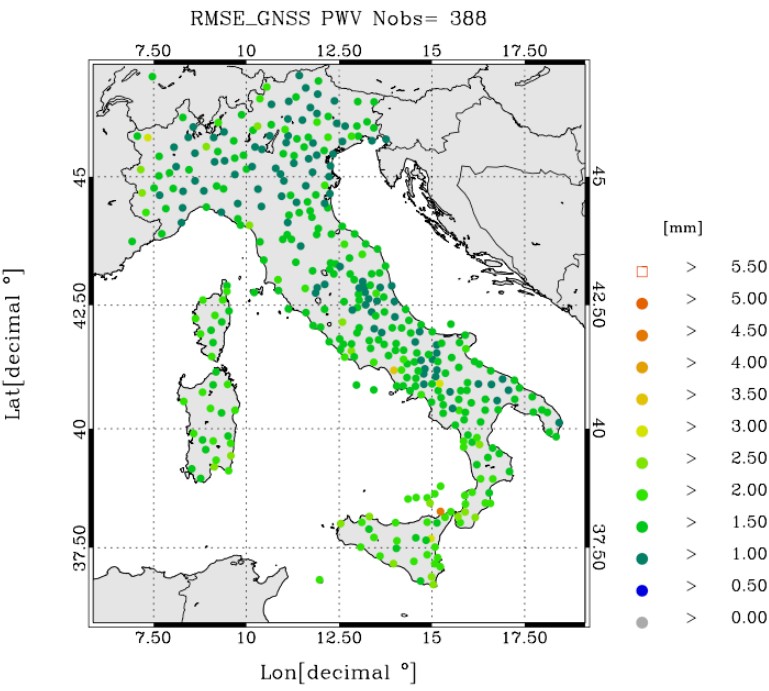

c)

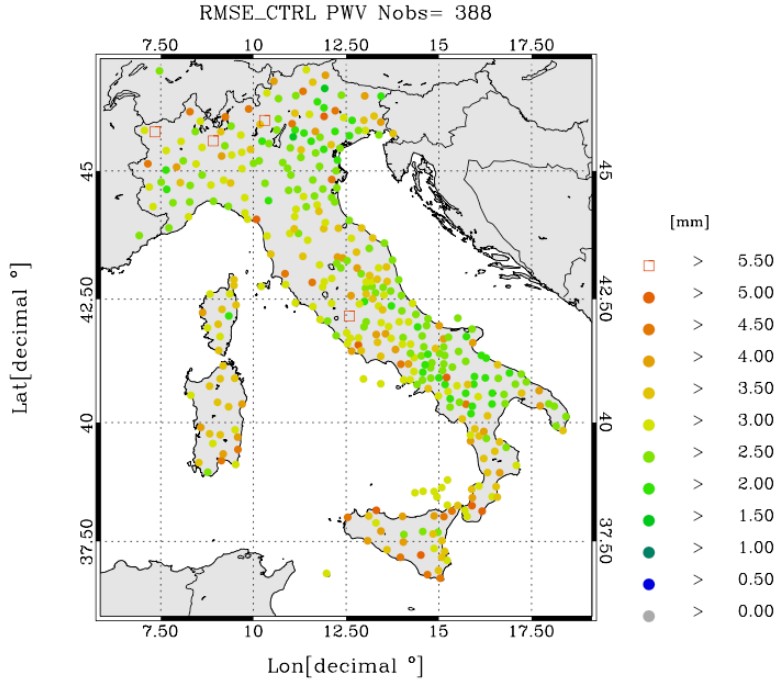

d)

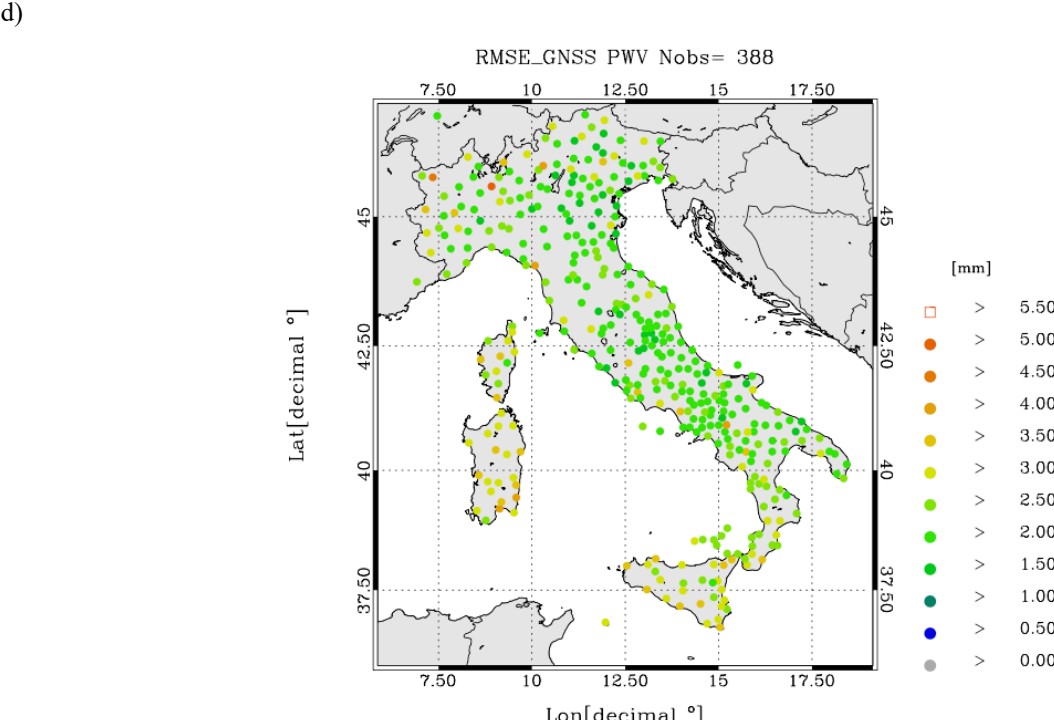

**Figure 8: Maps of the PWV RMSE for the first (a, b) and sixth (c, d) forecasting hour. Panels a) and c) are for CTRL simulations,**
**panels b) and d) are for GNSS simulations.**

### 3.4 Results for the precipitation forecast over the whole period

In this section we show the impact of GNSS-ZTD data assimilation on the precipitation forecast statistics for the whole period (2-31 October 2019). Two precipitation phases are considered: the first 3h of each simulation after the last assimilation time (also 0-3h) and the following 3h, i.e. from the 3rd to the 6th hour of each simulation after the last assimilation time (hereafter 3-6h), to evaluate for how long the GNSS-ZTD data assimilation impacts the precipitation forecast.

Performance diagrams are shown in Figure 9 for both CTRL and GNSS and for three different precipitation thresholds, namely 1 mm/3h, 10 mm/3h and 30 mm/3h. Starting from the analysis of the 0-3h forecast, the following four points can be noticed: a) the GNSS-ZTD assimilation improves the precipitation forecast for all precipitation thresholds as the GNSS symbols are always closer to the upper right corner compared to the CTRL; b) the FBIAS of the CTRL is underestimated for all precipitation thresholds; c) the FBIAS, POD, FAR and the TS of the GNSS are always larger than the corresponding CTRL; d) the performance of both GNSS and CTRL decreases with the increasing precipitation thresholds. The point a) shows that the precipitation forecast is improved by GNSS-ZTD data assimilation; the points b) and c) show that the assimilation of GNSS-ZTD improves the CTRL underestimation of the precipitation events for all thresholds. The point c) shows that, while the performance of GNSS is improved compared to CTRL, the added value of GNSS-ZTD data assimilation is reduced by the FAR increase. This point will be confirmed by the results of the statistical test shown later in this section. The point d) shows the well-known difficulties to correctly predict the correct amount, location, and timing of precipitation events as their intensity increases.

Results for the 3-6h phase are like those of the 0-3h period, highlighted in the four points above, with the exception that the FAR for the 30 mm/3h threshold is reduced by GNSS-ZTD data assimilation. In addition, the comparison of Figures 9a and 9b shows that the improvement of the model performance decreases with increasing forecasting time because the GNSS and CTRL symbols are closer for the 3-6h phase compared to the 0-3h forecast. This is in agreement with the analysis of the PWV presented in the previous section.

All in all, there are two important points to remark for the 0-3h and 3-6h performance diagrams: a) the performance of CTRL is improved by GNSS-ZTD data assimilation; b) the impact of assimilating GNSS-ZTD on the precipitation forecast lasts at least 6h.

a)

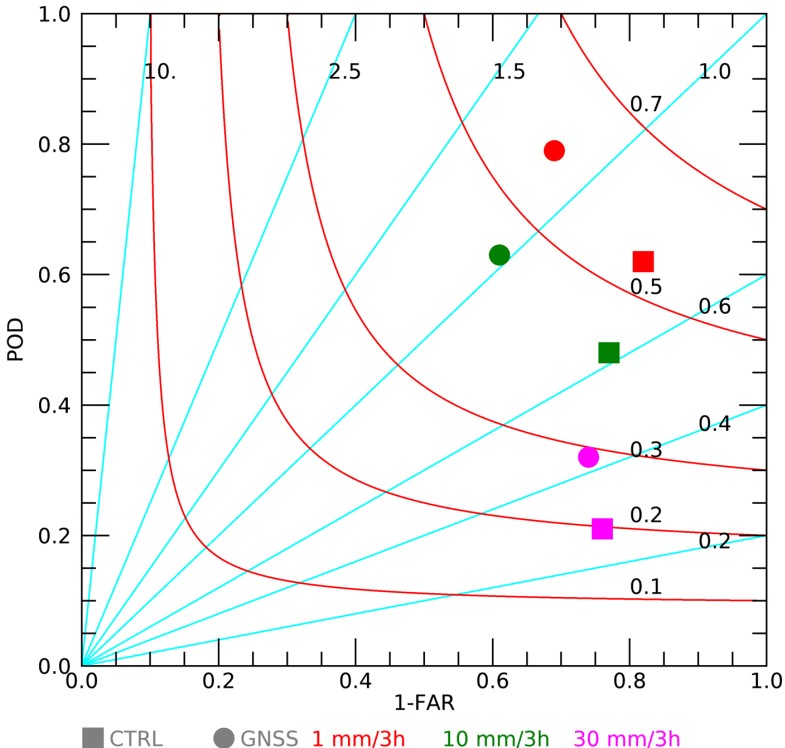

b)

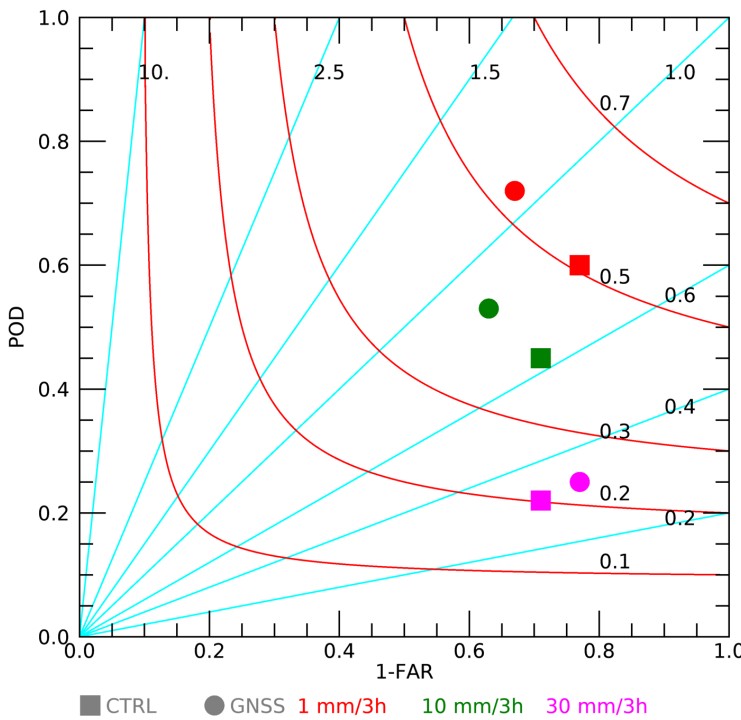

**Figure 9: Performance diagram showing precipitation scores over the whole period for CTRL (squares) and GNSS (circles) for the 0-3h period (a) and for the 3-6h period (b) after the assimilation phase for three different precipitation thresholds: 1 mm/3h (in red), 10 mm/3h (in green) and 30 mm/3h (in magenta).**

To better examine the difference of the statistical scores of the CTRL and GNSS forecasts shown in Figure 9, the results of a bootstrap statistical test (see Appendix B for details) are shown in Table 2 for the 0-3h and 3-6h forecast phases. The level of statistical significance is shown in plain text (90, 95, 99%) if the GNSS simulations perform better than CTRL, in bolded italic for the opposite case. The statistical test was computed for 1mm/3h, 5mm/3h and then every 5mm/3h up to 50 mm/3h thresholds. Results are shown up to 30 mm/3h thresholds because differences were not statistically significant for larger

precipitations. From Table 2, it is apparent that the assimilation of GNSS-ZTD has a good and significant impact on the precipitation forecast because FBIAS and POD are improved for several thresholds. The FAR is worsened by the data assimilation because of the larger number of false alarms predicted by GNSS compared to CTRL. This limits the improvement of ETS, which accounts for both hits and false alarms; however, the ETS has statistically significant improvements at 25 and 30 mm/3h showing a significant and positive impact of GNSS-ZTD data assimilation for moderate-intense precipitation

forecast. It is finally noticed the improvement of the FBIAS, POD, and ETS scores for both 0-3h and 3-6h phases, showing that the impact of GNSS-ZTD data assimilation lasts at least 6h into the forecast.

The positive impact of the GNSS-ZTD data assimilation on the rainfall forecast was explored with success in several studies (most of those cited in the Introduction) and the results of this paper quantify it in a robust way. This study, however, is limited

to the wet season when the synoptic scale forcing is well active (see the Supplemental material for a short discussion on the synoptic conditions during October 2019). Some papers (Giannaros et al., 2020; Boniface et al. 2009) show a larger impact of GNSS-ZTD data assimilation during the dry period, when the synoptic forcing is weaker, and this could be explored in future studies.

**Table 2: Results of the resampling statistical test. Statistical significance of 90, 95 and 99% is shown in plain-text when GNSS-ZTD data assimilation improves the statistics, while it is shown in bolded italic when the GNSS-ZTD data assimilation has a negative impact on the statistics. Significant values less than 90% are not shown. The first number in each cell refers to the first 3h of forecast (0-3h), while the second number refers to the second 3h of forecasts (3-6h).**

|       | 1mm/3h | 5mm/3h | 10mm/3h | 15mm/3h | 20mm/3h | 25mm/3h | 30mm/3h |
|-------|--------|--------|---------|---------|---------|---------|---------|
| FBIAS | 99;99  | 99;99  | 99;99   | 99;/    | 99;/    | 95;90   | 95;/    |
| ETS   | 90;/   | /;/    | /;/     | /;/     | /;/     | 90;90   | 95;/    |
| POD   | 99;99  | 99;99  | 99;99   | 99;/    | 99;/    | 90;90   | 95;/    |
| FAR   | *99;99* | *99;95* | *99;/*  | *95;/*  | *90;/*  | /;/     | /;/     |

4 Sensitivity tests

4.1 Data thinning experiment

As discussed in Section 2, the ZTD observation-error covariances are assumed to be uncorrelated in space and the **R** matrix is diagonal. However, the ZTD observation-errors are correlated to some degree and the assimilation of GNSS-ZTD data from receivers that are too close is sub-optimal. To consider in more detail this issue we did a sensitivity test for 16 days of October 2019 and data thinning. The days selected are those from 14 to 23 October that, as better discussed in the Supplemental material of this paper, were characterized by heavy and widespread rainfall especially in the NW of Italy and the days from 5 to 10 October, that were also characterized by moderate to intense rainfall. In this experiment (THIN), we first derive a distance where the observation errors become uncorrelated ($d$) and then GNSS-ZTD observations are assimilated only if their distance is larger than the $d$ value. To estimate this distance, we use the method of Desdrozier et al. (2015; see also Bennitt et al., 2017). In this method, observation minus background (innovation) and analysis minus observations (residual) statistics are used. To compute the observation-error decorrelation distance, GNSS-receivers are binned every 12.5 km and the covariance is computed as a function of the distance:

$$cov = \frac{1}{\sum_{i=1}^{n} m_i} \sum_{i=1}^{n} \sum_{i=1}^{m_i} [y_i - H_i(\boldsymbol{x_a})][y_j - H_j(\boldsymbol{x_b})] \tag{3}$$

Where $m_i$ is the number of GNSS receivers that fall in a specific bin at a certain distance from the $i$-th receiver, $n$ is the number of GNSS receivers, $H_i$ is the observation operator. By Eqn. (3) the errors covariance is computed as a function of the distance.

Another parameter needed to assess the observation-error correlation distance $d$ is the observation standard error variance, which is the covariance for the zero distance. It is computed as:

$$\sigma^2 = \frac{1}{n}\sum_{i=1}^{n}[y_i - H_i(x_a)][y_i - H_i(x_b)] \tag{4}$$

    Figure (10a) shows the behavior of the error covariance as a function of the distance (6.25 km, which is the central distance of
the first bin, 18.75 km, which is the distance of the second bin, etc.). The values of covariances have been normalized by the variance $\sigma^2$ (Eqn. 4). In statistics, correlation values below 0.2 are usually assumed as negligible and, from Figure 10a, the 20 km distance can be reasonably assumed as the distance for observation-error decorrelation. The assimilation of two receivers, whose distance is less than 20 km, is considered as sub optimal and is avoided in the THIN experiment.

    To obtain GNSS receivers with about 20 km spacing, we divided the domain (5°-20°E and 35°-50° N) with a regular longitude-
latitude lattice with 0.25° spacing and we selected one GNSS receiver for each grid box. If more receivers are present in the grid-box, the one closest to the grid box center is selected. The GNSS receivers used in the data thinning experiment are 259 (Figure 10b).

**a)**

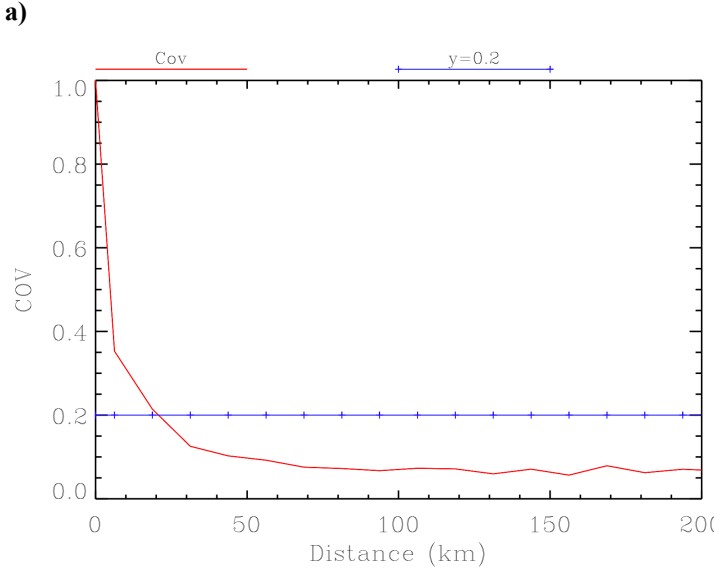

**B)**

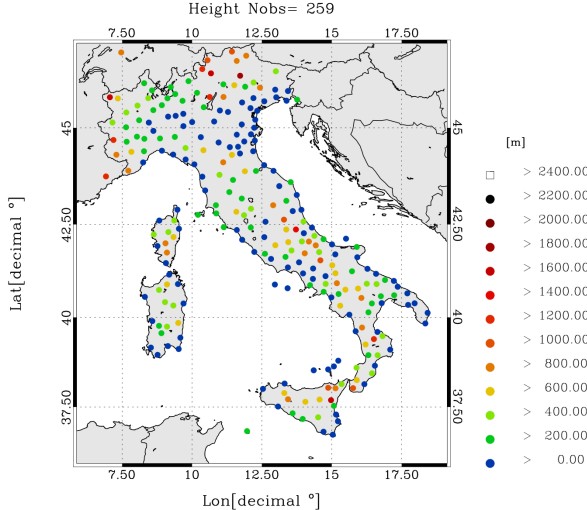

Figure 10: a) error covariance as a function of the distance (red curve). Covariances are normalized with the variance s². The y=0.2 curve is plotted for reference; b) GNSS receivers used for the forecast verification; c) GNSS receivers used for data assimilation.

Figure 11 shows the FBIAS, ETS, POD and FAR for CTRL, GNSS and THIN and BIAS experiments (the last is considered in the following section) for the 0-3h forecast phase. Here we limit the discussion to this phase as the differences between THIN and BIAS are lower for the 3-6h phase. Considering the four scores it is apparent that the difference between THIN and GNSS is small, especially for precipitation thresholds larger than 40 mm/3h. For lower rainfall thresholds the THIN experiment is slightly worse. All in all, these results don't show an improvement for data thinning experiment. This could be related to several factors and three of them seem more relevant: a) the limited period considered; b) the importance of the local scale in water vapour distribution; and c) the fact that the Desdrozier method is an estimate of the optimal distance between two receivers to be assimilated. So, considering the results of data thinning experiment, further research must be done to have a definitive answer.

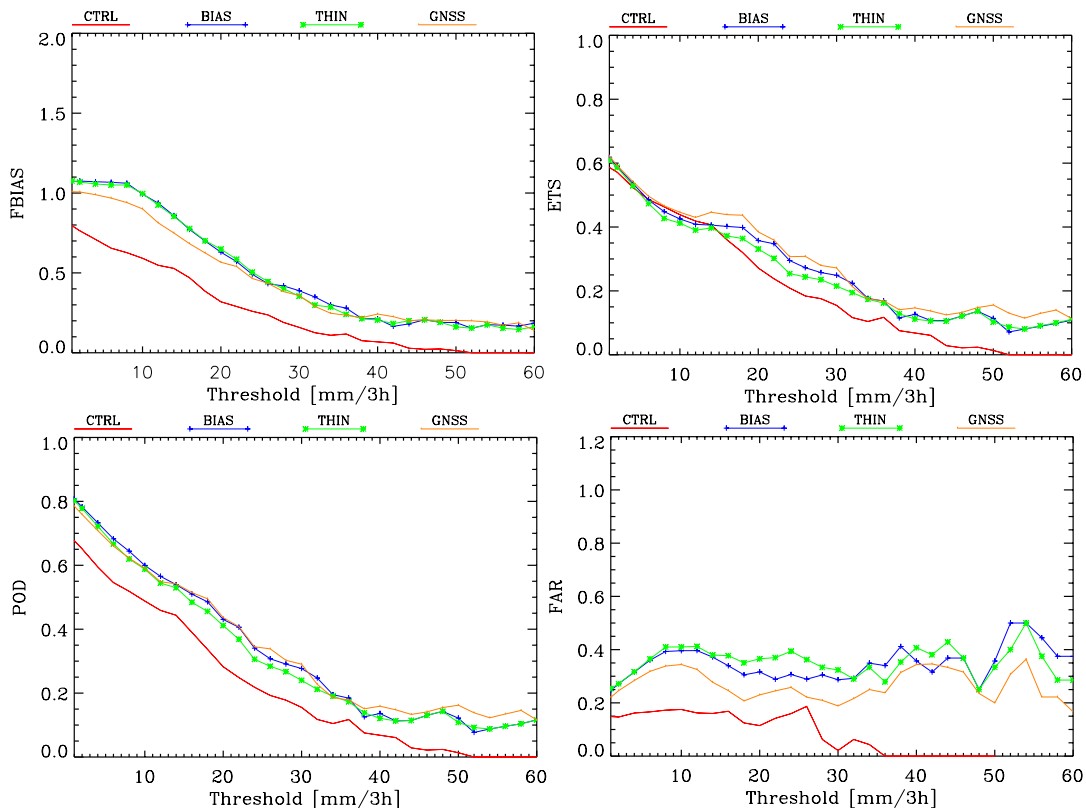

Figure 11: a) FBIAS for CTRL, GNSS, THIN and BIAS experiments; b) as in a) for ETS; c) as in a) for POD; d) as in a) for FAR. Scores are computed for the 1 mm/3h threshold and every 2mm/3h from 2  mm/3h to 60 mm/3h. Scores are computed for the 16 days between 5 and 10 and between 14 and 23 October 2019.

4.2 Impact of the bias removal

**It is interesting to study the impact of the bias removal on the precipitation forecast. The 3DVar assumes that observations are unbiased with respect to WRF model and the bias was removed before data assimilation as discussed in Section 2.1. This approach assumes that we know the observations for the forecasted period, which is not possible in an operational context. While other methods are possible (Bennitt and Jupp, 2017; Giannaros et al., 2020; Benjamin et al., 2016 among others), it is expected that in our case the impact of the bias removal is small because we limited the data assimilation to GNSS receivers with a maximum height difference with the model orography of 300 m and because WRFDA takes into account for the difference between the GNSS receiver and model height. To better assess this point we compared the results of simulations with or without bias removal for the sixteen days between 5 and 10 October**

**and between 14 and 23 October 2019. The experiment without bias removal is named BIAS (Figure 11). As expected, results show small differences between BIAS and GNSS experiments with the latter performing slightly better.**

## 4 Conclusions

In this paper we studied the impact of GNSS-ZTD data assimilation on the short-term (up to 6h) forecast over Italy for the month of October 2019, using the WRF model. A dense dataset of 388 GNSS receivers was used.

The comparison between first guess and ZTD observations showed that the forecast without data assimilation underestimates the water vapor content for the period. The GNSS-ZTD data assimilation partially compensates for this underestimation increasing the water vapor content in the atmosphere. The data assimilation roughly halves both the BIAS and the RMSE statistics for the ZTD. The analysis over the Italian territory shows a general reduction of BIAS and RMSE of the PWV thanks to the rather homogeneous and widespread coverage of GNSS receivers.

The case study on 15 October 2019 was chosen to show the impact of assimilating GNSS-ZTD on the precipitation forecast. The analysis at 18 UTC shows that 3DVar spreads the water vapor innovations both horizontally and vertically, and the improvement of the short-term precipitation forecast is notable, the main drawback being the increase of false alarms over Central Italy.

Considering the statistics over the whole period, the analysis of the PWV forecast shows a clear improvement for the simulations with data assimilation. This improvement is widespread over the Italian territory. The PWV RMSE is almost halved for the first forecasting time. As expected, the improvement of the PWV RMSE decreases with forecasting time as the effects caused by data assimilation are partially advected out of Italy.

Assimilating GNSS-ZTD increases the precipitation of the short-term forecast compared to CTRL, which shows a systematic underestimation of the FBIAS. Two 3h periods were considered after the last analysis time: 0-3h and 3-6h. For both periods, the FBIAS and the POD are increased by data assimilation and the control forecast is improved. As a drawback, the number of false alarms was increased by GNSS-ZTD data assimilation. ETS is also improved by data assimilation, even if the increase in the number of false alarms limits the impact of data assimilation on this score.

The results of the resampling statistical test show that the improvement for the FBIAS and POD are statistically significant for several thresholds up to 30 mm/3h. The FAR increase is statistically significant up to 20 mm/3h, while ETS has significant improvements for 1, 25 and 30 mm/3h.

We started to evaluate the impact of data thinning and of the bias removal in this work. The impact of bias removal is expected small because we assimilated data of GNSS receiver whose height difference with the model doesn't exceed 300 m. This was confirmed by the results of a numerical experiment spanning 16 heavy precipitation days. Also the data thinning experiment did not show consistent differences between the GNSS and THIN experiments. We believe that a longer numerical experiment must be considered to better assess this point.

All in all, the statistical analysis reveals a positive impact of GNSS-ZTD and the improvement is apparent up to 6h.

While the results of this paper are encouraging, there are several points that need future studies. First, the Mediterranean climate has an important seasonal variability and the impact of assimilating GNSS-ZTD must be studied in different seasons. Second, this study refers to only one month: longer periods must be considered to give a more robust assessment of the impact of GNSS-ZTD data assimilation on the forecast over Italy. Third, other techniques to assimilate GNSS data, as assimilating the precipitable water vapor (PWV) and the sensitivity of the results to the background error matrix should be considered in

future studies. Fourth, assimilating GNSS-ZTD in real-time over Italy and assimilating the ZTD gradients are two subjects that deserves detailed future research.

**Acknowledgments**

This work was realized in the project AEROMET (AERO spatial data assimilation for METeorological weather prediction)

funded by the Lazio region - FESR Fondo Europeo di Sviluppo Regionale Programma Operativo regionale del Lazio. ECMWF is acknowledged for providing the computational resources for this work.

**Appendix A: Precipitation scores**

Scores are computed starting from contingency tables (Table A1) for dichotomous events, i.e. events that can have only two values. In this case the two values are "yes" or "no" and the event is: ''precipitation is above or equal a certain threshold''.

**Table A1. Contingency table for dichotomous events.**

|  |  | Forecast | |
| --- | --- | --- | --- |
|  |  | Yes | No |
| Observation | Yes | $a$ | $c$ |
|  | No | $b$ | $d$ |

In Table A1, $a, b, c$ and $d$ have the following meaning:

- $a$ represents the hits. A hit occurs when both the precipitation forecast and the corresponding rain gauge observation

are above or equal to a rainfall threshold;

- $b$ represents the false alarms. A false alarm occurs when the precipitation forecast is above or equal to a rainfall threshold, while the corresponding rain gauge observation is below the same threshold;

- $c$ represents the misses, i.e. when the precipitation forecast is below a rainfall threshold, while the corresponding rain gauge observation is above or equal to the same threshold;

- $d$ represents the correct no forecasts, i.e. when both the precipitation forecast and the corresponding observation are below a rainfall threshold.

The raingauges used for computing the elements of the contingency tables are shown in Figure A1. Not all raingauges are reporting data for each 3h time interval considered and the average number of reports is 2700.

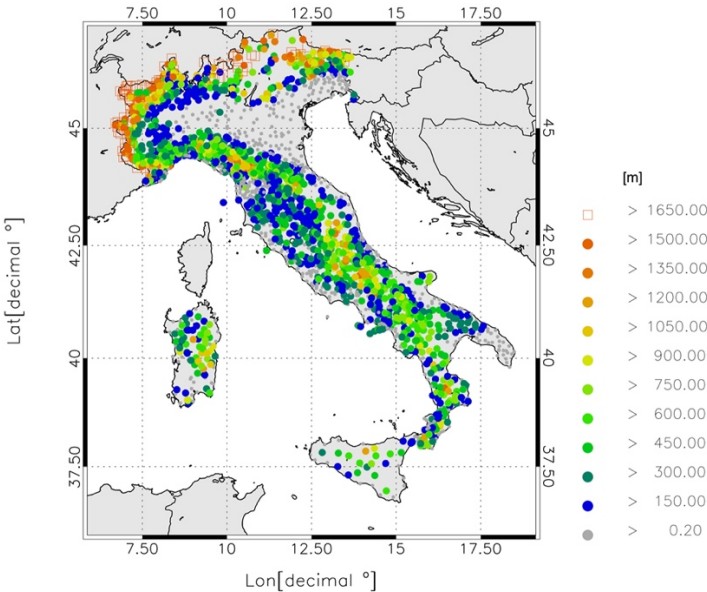

Figure A1: raingauges elevations [m].

Starting from the values in Table A1, the following scores can be defined:

- Frequency Bias (FBIAS)

$$FBIAS = \frac{a + b}{a + c}$$

which represents the frequency of the predicted events above a rainfall threshold with respect to observed events and can assume values between 0 and ∞, 1 being the best value possible.

- Probability of Detection (POD):

$$POD = \frac{a}{a + c}$$

which is the ratio between the number of correctly predicted events and the number of observed events and can assume values between 0 and 1, with 1 the best value possible.

- Threat Score (TS):

$$TS = \frac{a}{a + b + c}$$

which is given by the ratio between the number of events correctly predicted and the sum of observed and predicted events. TS assumes values between 0 and 1, and 1 is the best value that can be obtained.

- Equitable Threat Score (ETS):

$$ETS = \frac{a - a_r}{a + b + c - a_r}, \qquad \text{where } a_r = \frac{(a+b)(a+c)}{a+b+c+d}$$

ETS is similar to TS, but takes into account for the possibility of correctly forecasting an event by chance. It varies in the range between -1/3 and 1, being 1 the best value. A value of zero represents a useless forecast, in which the probability of correctly predicting an event is random.

- False Alarm Rate (FAR):

$$FAR = \frac{b}{a + b}$$

which is the ratio between false alarms and number of predicted events and can assume values between 0 and 1. In this case the best value possible is 0. Scores were computed after summing contingency table elements for each model over all simulations (4x30=120 times in this paper, both for the 0-3h and 3-6h forecast interval).

## Appendix B: The resampling method

The resampling method from Hamill (1999) is used for assessing if score differences are statistically significant in a confidence interval. For this purpose, an hypothesis test is performed. The null hypothesis is that the scores of the considered models do not differ.

In our case, the null hypothesis is that the differences between the two model scores, i.e. CTRL and GNSS, are zero. Let $S_1$ and $S_2$ be a generic score, namely FBIAS, ETS, POD and FAR (see Appendix 1 for details), for the two model types, CTRL and GNSS, respectively. The null hypothesis can be then written as:

$$H_0: S_1 - S_2 = 0$$

The test statistic is calculated after summing contingency table elements for each model over all simulations (4x30=120 times in this paper, both for the 0-3h and 3-6h forecast intervals). The contingency tables can be written as a vector:

$$x_{i,j} = (a, b, c, d)_{i,j}$$

Where $i$ is the model type (CTRL or GNSS) and $j$=1, …,120 is the contingency table for each time interval.

The contingency table elements are then summed over all times for both model forecasts CTRL and GNSS:

$$S_i = f\left(\sum_{j=1}^{n} x_{i,j}\right)$$

and the test statistic is given by the difference between S1 and S2.

Resampled sums can be done after randomly choosing either one or the other model for each time. Let $I_j$ be a random number which can assume values 1 or 2 for CTRL and GNSS, respectively, with $j$=1,…,120. We can then sum the shuffled vectors over all times:

$$S_1^* = f\left(\sum_{j=1}^{n} x_{Ij,j}\right)$$

and sum separately data not selected for the first sum, given by the index $(3 - I_j)$:

$$S_2^* = f\left(\sum_{j=1}^{n} x_{(3-Ij),j}\right).$$

Each sample is produced to be consistent with a null distribution, i.e. score differences $(S_1^* - S_2^*)$ are zero.

This random sampling is performed many times (10 000).

We consider the significance levels of α= 0.01, α= 0.05 and α= 0.1 and we test the null hypothesis $H_0$ with the percentiles. Let

$t_L = \frac{\alpha}{2}$ and $t_U = \frac{(1-\alpha)}{2}$.

The null hypothesis $H_0$ is rejected at the level 90 % (α=0.1), 95%(α=0.05) or 99% (α=0.01) if

$$S_1 - S_2 < t_L$$

or

$$S_1 - S_2 > t_U$$

where $S_1$ and $S_2$ are the non-resampled scores.

**Competing interests**

The contact author has declared that none of the authors has any competing interests.

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
