# Peer review of "The impact of GNSS Zenith Total Delay data assimilation on the short-term precipitable water vapor and precipitation forecast over Italy using the WRF model"

_Natural Hazards and Earth System Sciences, 2023_

## Referee Comment (RC1)

Review of "The impact of GNSS Zenith Total Delay data assimilation on the short-term precipitable water vapor and precipitation forecast over Italy using the WRF model" by Torcasion R.C., Mascitelli A., Eralini E., Barindelli S., Tagliaferro G., Puca S., Dietrich S. and Federico S. (Manuscript ID: NHESS-2023-18)

The current study examines the impact of assimilating GNSS-ZTD data on the performance of the WRF model in terms of simulating PWV and rainfall. The topic may be of interest, but it has been quite extensively addressed in previous studies, even in Italy (Lagasio et al., 2019; Mascitelli et al., 2019, 2021), which is the study area of the present paper. Unfortunately, the current study does not add something new to the existing literature, either with respect to the methodology nor concerning the results. A widely used (and relatively simple compared to 4Dvar, 3D-EnVar etc.) data assimilation (DA) system has been employed for performing very short-term DA experiments that cover the period from 2 to 31 October, 2019. In the framework of highlighting the novelty of the current study (Lines 82-85), this period is characterized as "longer" compared to previous studies. However, Mascitelli et al., (2019, 2021) also performed 1-2 month-long DA experiments. Further, the use of 388 GNSS receivers in the current study is highlighted, but Lagasio et al. (2019) also used 375 GNSS stations (in addition to satellite data) for their DA experiments. Based on the above, it is clear that the novelty of the present paper is lacking. I suggest the authors to reframe the conceptualization of their work considering the notes on future studies they suggest (Lines 419-425), as well as other pathways that can add value and novelty to their study.

Besides the above critical issue, other major and minor review points are highlighted below.

Major

- Study period: Please provide more information on the "moderate to intense precipitation events" that took place within October 2019 (dates, sums of precipitation, synoptic conditions etc.). This is important, because previous studies showed mixed ZTD DA impact, depending on the characteristics of the simulated events (e.g. synoptic-scale vs. convective).
- ZTD observations: National and regional networks were used for deriving the ZTD observations. Thus, a critical question arising is related to the accuracy of each network. This is important because the observational errors affect the DA process and the final outcomes. Please clarify if any accuracy assessment and pre-processing was performed for the ZTD observations and justify the selection of a fixed value of 5 mm as ZTD error for all networks.
- Lines 211-221: Please clarify why PWV is calculated using the observed ZTD and WRF-modeled ZHD. Is this computation corresponds to the forecasted PWV? Please clarify how the observational based PWV is computed.
- Case study analysis: The case study results are examined on the basis of maps comparison. Please provide a statistical evaluation of the results (as in the whole period analysis).
- Results and discussion: Please enrich this section in terms of interpreting the results and placing them in in the context of the related literature.

Minor

Overall, the paper is well written and structured, but some points can be improved. These include the frequent use of separate lines instead of longer paragraphs. For instance, the Abstract should be a single paragraph. Further,

- Abstract: Lines 23-25 (about the results) should be placed before Lines 21-22 (about the results). Further, it seems that there is repetition in Lines 21-22 and 28-30. Please be more specific concerning "showed an improvement of the precipitation forecast in different ways". Please revise "model 4.1.3" to "model, version 4.1.3,".
- Introduction: The first paragraph lacks a conceptual connection to the content provided in the next paragraphs. This is also true for the sentence in Lines 68-69 (please refer to the countries) in relation to the previous paragraph. Please make clear that the studies of Lagasio et al. (2019) and Mascitelli et al. (2019, 2021) were performed over Italy (Lines 76-81). Please provide abbreviations for the terms 3DVar, 4DVar etc.
- Lines 112-113: The "background simulations" usually refer to those for deriving the model background errors for DA. Thus, I suggest renaming the experiments without DA to "control simulations".
- Please provide a map in Appendix A showing the locations of the rain gauges used for evaluating the model results.
- Please specify the reason of examining the innovations during the case study analysis (Lines 248-262). To my understanding this is done to highlight that the ZTD DA assimilation leads to actual modeled differences related to PWV that are not random.
- Figure 5: Please indicate the axis of the cross-section.

References

Lagasio, M., Parodi, A., Pulvirenti, L., Meroni, A.N., Boni, G., Pierdicca, N., Marzano, F.S., Luini, L., Venuti, G., Realini, E.,Gatti, A., Tagliaferro, G., Barindelli, S., Monti Guarnieri, A., Goga, K., Terzo, O., Rucci, A., Passera, E., Kranzlmueller, D., Rommen, B.: A Synergistic Use of a High-Resolution Numerical Weather Prediction Model and High-Resolution Earth Observation Products to Improve Precipitation Forecast. Remote Sens.-Basel, 11, 2387. https://doi.org/10.3390/rs11202387, 2019.

Mascitelli, A., Federico, S., Fortunato, M., Avolio, E., Torcasio, R. C., Realini, E., Mazzoni, A., Transerici, C., Crespi, M., Dietrich, S.: Data assimilation of GNSS-ZTD into the RAMS model through 3D-Var: preliminary results at the regional scale. Meas. Sci. Technol. 30, 055801 (14pp). https://doi.org/10.1088/1361-6501/ab0b87, 2019.

Mascitelli, A., Federico, S., Torcasio, R. C., Dietrich, S.: Assimilation of GPS Zenith Total Delay estimates in RAMS NWP model: Impact studies over central Italy. Adv. Space Res., https://doi.org/10.1016/j.asr.2020.08.031, 68, 12, pp 4783-4793, 2021.

---

## Author Comment (AC1)

Review of "The impact of GNSS Zenith Total Delay data assimilation on the short-term precipitable water vapor and precipitation forecast over Italy using the WRF model" by Torcasio R.C., Mascitelli A., Realini E., Barindelli S., Tagliaferro G., Puca S., Dietrich S. and Federico S. (Manuscript ID: NHESS-2023-18)

First of all, we acknowledge the reviewer for the careful review of the paper. In the following our answers are in black and the reviewer comments are in blue. Before entering the discussion of the major and minor points, we answer to the first part of the reviewer comment, that is defined as critical by the reviewer. This answer should clarify better the novel aspects of this paper compared to previous papers over Italy.

The current study examines the impact of assimilating GNSS-ZTD data on the performance of the WRF model in terms of simulating PWV and rainfall. The topic may be of interest, but it has been quite extensively addressed in previous studies, even in Italy (Lagasio et al., 2019; Mascitelli et al., 2019, 2021), which is the study area of the present paper. Unfortunately, the current study does not add something new to the existing literature, either with respect to the methodology nor concerning the results.

We cannot agree with this comment. In this paper, for the first time over Italy, this amount of GNSS-ZTD data, the convective allowing horizontal resolution (3km), and the period of one month (we choose October because it is a rainy month over Italy) in Very Short Term forecast configuration (120 simulations for each type) are used to assess the impact of GNSS-ZTD data assimilation on the precipitation forecast. The impact of GNSS-ZTD data assimilation on the precipitation forecast has not been assessed thoroughly over Italy and work is still to be done (for example to assess the impact of GNSS-ZTD data assimilation on the precipitation forecast in different seasons). Considering the comparison with other studies around the world, this paper is a first step to fill the gap between what was done over Italy and other countries. We clarify better the comparison with existing literature over Italy below.

A widely used (and relatively simple compared to 4Dvar, 3D-EnVar etc.) data assimilation (DA) system has been employed for performing very short-term DA experiments that cover the period from 2 to 31 October, 2019.

The choice of 3D-Var is motivated by the fact that this data assimilation method is used in the agreement between the CNR-ISAC and the Department of Civil Protection (DPC) of Italy. In this agreement, CNR-ISAC provides to DPC a forecast that has the same configuration used in this paper (Very short-term forecast approach, i.e. 6 h of data assimilation is followed by 6h of forecast and four forecasts are issued for each day), with the same data assimilation scheme. In this agreement we assimilate radar reflectivity by 3D-Var and lightning by nudging, but not GNSS-ZTD. So, in this experiment we use same configuration as in the agreement with the DPC to evaluate the potential of GNSS-ZTD data assimilation in the operational context..

In the framework of highlighting the novelty of the current study (Lines 82-85), this period is characterized as "longer" compared to previous studies. However, Mascitelli et al., (2019, 2021) also performed 1-2 month-long DA experiments. Further, the use of 388 GNSS receivers in the current study is highlighted, but Lagasio et al. (2019) also used 375 GNSS stations (in addition to satellite data) for their DA experiments. Based on the above, it is clear that the novelty of the present paper is lacking. I suggest the authors to reframe the conceptualization of their work considering the notes on future studies they suggest (Lines 419-425), as well as other pathways that can add value and novelty to their study.

We know the three papers cited by the reviewer and these works were cited already in our paper. In addition, many of the authors of this paper also co-authored the papers Mascitelli et al., 2019 and 2021 and few authors of this paper co-authored the Lagasio et al., 2019 paper.

Considering the paper of Mascitelli et al. (2019) "only" 26 geodetic receivers and three single frequency receivers were used for data assimilation in the RAMS@ISAC model. All these receivers were over Lazio region (Central Italy). GNSS-ZTD was assimilated for more than one month using 3DVar however, because of the low number of receivers and because of the intermittent character of the precipitation, only the impact of GNSS-ZTD data assimilation on the integrated water vapor forecast was considered. In the conclusions of

their paper the authors state: "The impact of the GPS-ZTD data assimilation must be studied for other parameters than IWV, in particular for precipitation."

Similarly, the paper of Mascitelli et al. (2021) considers 46 GNSS-ZTD receivers over central Italy (Lazio, Abruzzo and Sardinia regions). Data assimilation is performed by nudging for one month and the variable assimilated is the integrated water vapor (IVW). The analysis of both water vapor and precipitation is considered only for the assimilation phase, i.e. in the experiment considered in Mascitelli et al. (2021) IWV data are continuously assimilated without considering the forecast phase. Again, from their conclusions: "It is noted that the application shown in this paper does not consider the impact of GPS-ZTD assimilation on the precipitation forecast, but only on the simulation of the precipitation field (during the analysis phase). Future studies will consider the impact of GPS-ZTD data assimilation on RAMS@ISAC precipitation forecast.".

The paper of Lagasio et al. (2019) is very interesting because it assesses the impact of GNSS-ZTD data assimilation on the precipitation forecast but for two cases (Silvi Marina and Livorno). So, the sentence that we put into the paper about the novelty aspects and that is reported here "*This paper goes in a similar direction in the sense that it uses the GNSS-ZTD data assimilation to improve the precipitation and water vapor forecast over Italy. Compared to similar studies, however, it uses a longer period and/or a larger number of GNSS receivers widespread over the country, giving a robust assessment on the impact that GNSS-ZTD data assimilation can have on the forecast at the local scale.*" seems to us valid.

However we agree with the reviewer comment that something could be added to the paper in the direction of Lines 419-425. For this purpose we will add the results of a data thinning experiment. In this experiment a subset of the GNSS receivers is used instead of the 388 receivers used in the paper. Due to the limited time, the data thinning experiment is focused on the 10 days of the month (14-23 October 2019). These days were chacterized by intense precipitation over the NW of Italy with damages to infrastructures.

Importantly, we note an error in writing the paper: in discussing the results of Mascitelli et al (2021) we will change the sentence: "In both cases the assimilation showed a significant improvement in the short-term prediction of water vapor with smaller impact on the precipitation **forecast**." to "In both cases the assimilation showed a significant improvement in the simulation of water vapor with smaller impact on the precipitation simulation **during the assimilation period**."

Besides the above critical issue, other major and minor review points are highlighted below.

Major points

- Study period: Please provide more information on the "moderate to intense precipitation events" that took place within October 2019 (dates, sums of precipitation, synoptic conditions etc.). This is important, because previous studies showed mixed ZTD DA impact, depending on the characteristics of the simulated events (e.g. synoptic-scale vs. convective).

We will discuss the precipitation for the month of October 2019, better defining the number of rainy days (all 30 days, i.e. 2-31 October 2019, were rainy if we consider the whole Italian territory) the precipitation for October 2019 and its comparison with the last 11 years of data (for these years a comparable number of raingauges, as those used in the paper, is available). Some discussion about the synoptic conditions will be added. We will give a short summary of these results in the paper but, because it is a considerable amount of material, we will add it as supplement of the paper. To have an idea about the number of events considered in the paper we show here a table with the number of rainy events for different precipitation classes.

Table 1: Number and distribution of the rainfall events for 2019 and for 2012-2022.

| Rainfall Threshold (mm/3h) | Number - 2019 | Fraction - 2019 | Fraction – 11 years |
|---|---|---|---|
| 0-1 | 5316582 | 93.63 | 92.04 |

| 1-5 | 284420 | 3.87 | 4.92 |
|------|--------|------|------|
| 5-10 | 96070 | 1.29 | 1.66 |
| 10-20 | 56509 | 0.84 | 0.98 |
| 20-30 | 13872 | 0.22 | 0.24 |
| 30-50 | 6782 | 0.11 | 0.12 |
| 50-70 | 1522 | 0.02 | 0.02 |
| > 70 | 889 | 0.02 | 0.02 |

In the above table the fractions are computed compared to the total number of reports for 2019 (662759) and to the total number of events in the database for the last 11 years (5316582). Compared to the last 11 years the rainfall distribution for 2019 shows a larger fraction of events for very small or no rainfall (0-1 mm/3h) class, while it has lower fraction of events from 1-5 mm/3h up to 10-20 mm/3h. For larger thresholds the fraction of events occurred in 2019 is like the last 11 years. In any case the number of events considered in this work is high. For example, the events with rainfall larger than 30 mm/3h (i.e. the most intense threshold showing a significant difference between the control and the forecast using GNSS-ZTD data assimilation) are 9193.

- ZTD observations: National and regional networks were used for deriving the ZTD observations. Thus, a critical question arising is related to the accuracy of each network. This is important because the observational errors affect the DA process and the final outcomes. Please clarify if any accuracy assessment and pre-processing was performed for the ZTD observations and justify the selection of a fixed value of 5 mm as ZTD error for all networks.

   Even though different regional networks are considered in this paper, to reach the considerable number of GNSS-ZTD receivers used, the software and the processing method is the same for all the receivers. The GNSS-ZTD time series were visually checked and no specific differences among network arose. This justifies the choice of a constant error. The value of 5 mm was not specifically computed for this experiment but comes from previous comparisons that, in any case, do not extend to the whole Italy (Tagliaferro, 2021; Krietemeyer et al. 2018.; Mascitelli et al. 2019 and 2021). In these works, the GNSS-ZTD retrieved with the method used in our paper was compared with other methods and with radiosoundings. In general comparison with radiososndes shows differences in the range 1.0-1.5 cm (i.e. larger than the error used in this paper), while differences with other methods show differences between 0.1 and 0.8 cm. Now, the comparison with radiosondes is less representative of the GNSS-ZTD error because radiosondes can move far from the GNSS receiver, while the 0.5 cm used in this paper come from the comparison of the method used in this paper to estimate ZTD with other methods. A paragraph will be added to the paper to explain this point. Thanks for noting it.

- Lines 211-221: Please clarify why PWV is calculated using the observed ZTD and WRF-modeled ZHD. Is this computation corresponds to the forecasted PWV? Please clarify how the observational based PWV is computed.

We don't know the values of surface pressure and temperature for each receiver, and we use the output of the control model to get these values. Then we use the Saastamoinen (1972) formula to estimate ZHD, as in other papers (Rohm et al., 2019, for example). The observed PWV is retrieved from the ZTD observation and ZHD estimated by the WRF model. The modeled PWV is retrieved using both ZTD and ZHD from the WRF model. We will clarify this point in the revised version of the paper. The method used by WRF to compute ZTD is clearly explained in Rohm et al. (2019). We will use this reference. Thanks for noting this point.

- Case study analysis: The case study results are examined on the basis of maps comparison. Please provide a statistical evaluation of the results (as in the whole period analysis).

   Yes, we agree with the reviewer. We will put the precipitation score for the case study in the supplemental material of the paper. Instead of the performance diagrams, we will show the FBIAS,

ETS, POD and FAR for different precipitation threshold (1mm/3h and from 2 to 60 mm/3h every 2mm/3h). The example for the POD is shown below.

[Figure]

Figure: POD score for the rainfall forecast between 18 and 21 UTC on 15 October 2019.

- Results and discussion: Please enrich this section in terms of interpreting the results and placing them in in the context of the related literature.

Ok. We will enrich the discussion.

Minor points

Overall, the paper is well written and structured, but some points can be improved. These include the frequent use of separate lines instead of longer paragraphs. For instance, the Abstract should be a single paragraph. Further,

- Abstract: Lines 23-25 (about the results) should be placed before Lines 21-22 (about the results). Further, it seems that there is repetition in Lines 21-22 and 28-30. Please be more specific concerning "showed an improvement of the precipitation forecast in different ways". Please revise "model 4.1.3" to "model, version 4.1.3,".

OK.

- Introduction: The first paragraph lacks a conceptual connection to the content provided in the next paragraphs. This is also true for the sentence in Lines 68-69 (please refer to the countries) in relation to the previous paragraph. Please make clear that the studies of Lagasio et al. (2019) and Mascitelli et al. (2019, 2021) were performed over Italy (Lines 76-81). Please provide abbreviations for the terms 3DVar, 4DVar etc.

OK

- Lines 112-113: The "background simulations" usually refer to those for deriving the model background errors for DA. Thus, I suggest renaming the experiments without DA to "control simulations".

Ok, we will implement this naming.

- Please provide a map in Appendix A showing the locations of the rain gauges used for evaluating the model results.

We will add the following figure:

[Figure]

Figure: Elevations of the raingauges.

- Please specify the reason of examining the innovations during the case study analysis (Lines 248-262). To my understanding this is done to highlight that the ZTD DA assimilation leads to actual modeled differences related to PWV that are not random.

The reason for showing the innovation for the case study is that the difference between the precipitation with or without data assimilation can be well interpreted considering these innovations. A better explanation will be given in the paper.

- Figure 5: Please indicate the axis of the cross-section.

We will use the following figure:

[Figure]

References

Lagasio, M., Parodi, A., Pulvirenti, L., Meroni, A.N., Boni, G., Pierdicca, N., Marzano, F.S., Luini, L., Venuti, G., Realini, E.,Gatti, A., Tagliaferro, G., Barindelli, S., Monti Guarnieri, A., Goga, K., Terzo, O., Rucci, A., Passera, E., Kranzlmueller, D., Rommen, B.: A Synergistic Use of a High-Resolution Numerical Weather Prediction Model and High-Resolution Earth Observation Products to Improve Precipitation Forecast. Remote Sens.-Basel, 11, 2387. https://doi.org/10.3390/rs11202387, 2019.

Mascitelli, A., Federico, S., Fortunato, M., Avolio, E., Torcasio, R. C., Realini, E., Mazzoni, A., Transerici, C., Crespi, M., Dietrich, S.: Data assimilation of GNSS-ZTD into the RAMS model through 3D-Var: preliminary results at the regional scale. Meas. Sci. Technol. 30, 055801 (14pp). https://doi.org/10.1088/1361-6501/ab0b87, 2019.

Mascitelli, A., Federico, S., Torcasio, R. C., Dietrich, S.: Assimilation of GPS Zenith Total Delay estimates in RAMS NWP model: Impact studies over central Italy. Adv. Space Res., https://doi.org/10.1016/j.asr.2020.08.031, 68, 12, pp 4783-4793, 2021.

**References (will be added to the paper)**

Krietemeyer, A.; Ten Veldhuis, M.-c.; Van der Marel, H.; Realini, E.; Van de Giesen, N. Potential of Cost-Efficient Single Frequency GNSS Receivers for Water Vapor Monitoring. *Remote Sens.* 2018, *10*, 1493. https://doi.org/10.3390/rs10091493

Tagliaferro G.: On the Development of a General Undifferenced Uncombined Adjustment for GNSS, PhD Dissertation 2021, Politecnico di Milano, Department of Civil and Environmental Engineering.

---

## Author Comment (AC2)

Answers to reviewer n°2

The paper studies the impact of GNSS data assimilation over Italy with a focus on a 6-hour forecast based on precipitable water vapor and precipitation. It is evident from the study that GNSS DA improves the underestimation of water vapor by WRF.

We acknowledge the reviewer for reviewing the paper and for the useful comments.

I mostly agree with the points mentioned by **Referee #1**, however, I think that reframing the aim of the paper and adding some additional analysis that could bring some novelty to the paper. From the reply comments of the author to the reviewer, I see the author has already started to prepare results in the direction of data thinning experiments which is good.

We will discuss the results of the data thinning experiment and we will reframe the sentence about the novelty of the paper.

My major concern to add is that the data assimilation experiments need some more elaboration:

- regarding the observations assimilated in the BCKG experiment.

The BCKG experiment uses only initial and boundary conditions from ECMWF. No other data are assimilated to focus on the added value of GNSS-ZTD data assimilation alone. We will mention this explicitly into the paper. Thanks.

- regarding the GNSS data used for assimilation and if bias correction was performed before the assimilation of GNSS data.

Thank you for this comment, which gives the opportunity to clarify the point. We applied the bias corrections before data assimilation using the following procedure. First the raw GNSS data are assimilated in the 3DVAR to calculate the corrections that come from the background. The difference between the observation and the background is saved for each receiver and for each time giving the quantity $(O\text{-}B)_{i,t}$, where $i$ is the receiver index and $t$ is the time. The quantity $(O\text{-}B)_{i,t}$ takes into account for the difference between the model orography and receiver height that, in our case, is never larger than 300 m. For each receiver we compute the background bias by averaging $(O\text{-}B)_{i,t}$ over all times:

$$\overline{(O-B)}_i = \sum_{t=1}^{N} \frac{(O-B)_{i,t}}{N}$$

Where $N$ is the total number of times (i.e. observations) available for each GNSS receiver. Then we use the corrected observation $O'_{i,t}$ in the 3D-Var:

$$O'_{i,t} = O_{i,t} - \overline{(O-B)}_i$$

A paragraph will be added to the paper describing the procedure above. Similar methods were used in the bibliography.

My minor concern to add is about the tense usage in the text throughout the journal should be uniform. Also, the paragraph construction should be more refined regarding the main point to express in a context.

We will review the tense usage.

---

## Author Response (AR1)

Answers to the Reviewer #1 comments

Review of "The impact of GNSS Zenith Total Delay data assimilation on the short-term precipitable water vapor and precipitation forecast over Italy using the WRF model" by Torcasio R.C., Mascitelli A., Realini E., Barindelli S., Tagliaferro G., Puca S., Dietrich S. and Federico S. (Manuscript ID: NHESS-2023-18)

First of all, we acknowledge the reviewer for the careful review of the paper. In the following our answers are in black and the reviewer comments are in blue. The parts of the paper are referred in italic.

The current study examines the impact of assimilating GNSS-ZTD data on the performance of the WRF model in terms of simulating PWV and rainfall. The topic may be of interest, but it has been quite extensively addressed in previous studies, even in Italy (Lagasio et al., 2019; Mascitelli et al., 2019, 2021), which is the study area of the present paper. Unfortunately, the current study does not add something new to the existing literature, either with respect to the methodology nor concerning the results.

A widely used (and relatively simple compared to 4Dvar, 3D-EnVar etc.) data assimilation (DA) system has been employed for performing very short-term DA experiments that cover the period from 2 to 31 October, 2019.

In the framework of highlighting the novelty of the current study (Lines 82-85), this period is characterized as "longer" compared to previous studies. However, Mascitelli et al., (2019, 2021) also performed 1-2 month-long DA experiments. Further, the use of 388 GNSS receivers in the current study is highlighted, but Lagasio et al. (2019) also used 375 GNSS stations (in addition to satellite data) for their DA experiments. Based on the above, it is clear that the novelty of the present paper is lacking. I suggest the authors to reframe the conceptualization of their work considering the notes on future studies they suggest (Lines 419-425), as well as other pathways that can add value and novelty to their study.

We already answered to the above points in the open discussion and do not completely agree with the above considerations. In any case, because also the second reviewer posed the problem of the novelty of this paper we adjusted the sentence to put this paper better into the context. We wrote:

*"This paper goes in a similar direction in the sense that it uses the GNSS-ZTD data assimilation to improve the precipitation and water vapor forecast over Italy. It uses a period of one month (October 2019) and the data of 388 GNSS receivers widespread over the country, giving a robust assessment on the impact that GNSS-ZTD data assimilation can have on the forecast at the local scale."*

However, we agree with the reviewers comment that something could be added to the paper in the direction of Lines 419-425 to add same new aspects to the paper. For this purpose, we added the results of a data thinning experiment and of a bias removal experiment. Due to the limited computing resources, these experiments are focused on 16 precipitating days over the NW of Italy (from 5 to 10 and from 14 to 23 October 2019). See the supplemental material that we added to the paper for the synoptic characteristics of the days.

For the data thinning experiment, we applied the method of Desdrozier (see the reference into the revised paper) to quantify the distance where the observation error becomes uncorrelated (about 20 km in our case). Then we used a regular longitude-latitude lattice to select the receivers with a relative distance of 20 km. This procedure left 259 receivers. Then we repeated the experiment for the from 5 to 10 and from 14 to 23 October using only those 259 receivers. The results of this experiment show a small negative impact of the data thinning experiment. However, this result should be considered with caution because of the short period of time considered. It should be also considered that the optimal setting of 3DVar, with 20 km distance among receivers, could miss some local water vapour effects and this could have determined the small decrease of the performance when data thinning is applied.

As the observations that enters the 3DVar are considered unbiased compared to WRF, the ZTD observations that we used in the original experiment underwent a bias removal for the whole period (the procedure is now explained in Section 2.1). However, this method cannot be applied in the operational setting because it requires the knowledge of observations for future times. So, we studied the sensitivity of the results to the bias removal, simulating the sub period of 16 days using raw GNSS-ZTD, i.e. without bias removal. This sensitivity is expected to be small because we consider the assimilation of GNSS-ZTD only if the receiver

has a height difference with the WRF orography less than 300m and because the WRFDA takes into account for the difference between the receiver height and WRF orography. This is confirmed by the results of the bias removal experiment. The results of both experiments, i.e. data thinning and bias removal, are shown in Section 4.1 and 4.2, respectively. In the introduction, in order to refer to these experiments, we wrote:

"*In addition, the sensitivity of the results to the number of GNSS receivers used (data thinning) and to the bias correction is shown for a sub period (from 5 to 10 and from 14 to 23 October 2019).*"

Importantly, we note an error in writing the paper: in discussing the results of Mascitelli et al (2021) we chenged the sentence: "In both cases the assimilation showed a significant improvement in the short-term prediction of water vapor with smaller impact on the precipitation **forecast**." to "In both cases the assimilation showed a significant improvement in the short-term forecast of water vapor with smaller impact on the precipitation simulation **during the assimilation period**."

Besides the above critical issue, other major and minor review points are highlighted below.

Major points

- Study period: Please provide more information on the "moderate to intense precipitation events" that took place within October 2019 (dates, sums of precipitation, synoptic conditions etc.). This is important, because previous studies showed mixed ZTD DA impact, depending on the characteristics of the simulated events (e.g. synoptic-scale vs. convective).

We did a supplement of the paper in which we have discussed the precipitation for the month of October 2019, better defining the number of rainy days (all 30 days were rainy if we consider the whole Italian territory) the precipitation for the whole month  and its comparison with the last 11 years of data (for these years a comparable number of raingauges, as those used in the paper, is available). Some discussion about the synoptic conditions was also added based on the simulations of the WRF model. We recalled the supplemental material where/when necessary. To have an idea about the number of events considered in the paper we show here a table with the number of rainy events for different precipitation classes. We also show the rain distributions of the ERA5 dataset.

Table 1: Number and distribution of the rainfall events for 2019 and for 2012-2022.

| Rainfall Threshold (mm/3h) | Number - 2019 | Fraction - 2019 | Fraction – 11 years |
|---|---|---|---|
| 0-1 | 620521 | 93.63 | 92.04 |
| 1-5 | 25659 | 3.87 | 4.92 |
| 5-10 | 8520 | 1.29 | 1.66 |
| 10-20 | 5559 | 0.84 | 0.98 |
| 20-30 | 1489 | 0.22 | 0.24 |
| 30-50 | 749 | 0.11 | 0.12 |
| 50-70 | 153 | 0.02 | 0.02 |
| > 70 | 110 | 0.02 | 0.02 |

In the above table the fractions are computed compared to the total number of reports for 2019 (662760) and to the total number of events in the database for the last 11 years (5316582). Compared to the last 11 years the rainfall distribution for 2019 shows a larger fraction of events for very small or no rainfall (0-1 mm/3h) class, while it has lower fraction of events from 1-5 mm/3h up to 10-20 mm/3h. For larger thresholds the fraction of events occurred in 2019 is like the last 11 years. In any case the number of events considered in this work is high. For example, the events with rainfall larger than 30 mm/3h (i.e. the most intense threshold showing a significant difference between the control and the forecast using GNSS-ZTD data assimilation) are larger than 1000.

- ZTD observations: National and regional networks were used for deriving the ZTD observations. Thus, a critical question arising is related to the accuracy of each network. This is important because

the observational errors affect the DA process and the final outcomes. Please clarify if any accuracy assessment and pre-processing was performed for the ZTD observations and justify the selection of a fixed value of 5 mm as ZTD error for all networks.

Even though different regional networks are considered in this paper, to reach the considerable number of GNSS-ZTD receivers used, the software and the processing method is the same for all the receivers. The GNSS-ZTD time series were visually checked and no specific differences among network arose. This justifies the choice of a constant error. The value of 5 mm was not specifically computed for this experiment but comes from previous comparisons that, in any case, do not extend to the whole Italy (Tagliaferro, 2021; Krietemeyer et al. 2018.; Mascitelli et al. 2019 and 2021). Thanks for noting this important point. We wrote:

*"Different regional networks are considered to reach the considerable number of GNSS-ZTD receivers used in this paper; however, we used a constant value for the errors of all receivers because the software and the processing method are the same for all the receivers. Also, the GNSS-ZTD time series were visually checked and no specific differences among networks arose. The value of 5 mm was not specifically computed for this experiment but comes from previous comparisons that, in any case, do not extend to the whole Italy (Tagliaferro, 2021; Krietemeyer et al. 2018.; Mascitelli et al. 2019 and 2021). In these works, the GNSS-ZTD retrieved with the method used in our paper was compared with other methods and with radiosoundings. In general, comparison with radiosondes shows differences in the range 1.0-1.5 cm (i.e. larger than the error used in this paper), while comparison with other methods shows differences between 0.1 and 0.8 mm. The comparison with radiosondes is less representative of the GNSS-ZTD error because radiosondes can move far from the GNSS receiver, and the 0.5 cm used in this paper comes from the comparison with other methods to estimate ZTD. However, future experiments considering different errors for different networks should be done to assess more in detail this point. "*

• Lines 211-221: Please clarify why PWV is calculated using the observed ZTD and WRF-modeled ZHD. Is this computation corresponds to the forecasted PWV? Please clarify how the observational based PWV is computed.

We don't know the values of surface pressure and temperature for each receiver, and we use the output of the control model to get these values. Then we use the Saastamoinen (1972) formula to estimate ZHD, as in other papers (Rohm et al., 2019, for example). The observed PWV is retrieved from the ZTD observation and ZHD estimated by the WRF model. The modelled PWV is retrieved using both ZTD and ZHD from the WRF model. The method used by WRF to compute ZTD is clearly explained in Rohm et al. (2019). We will use this reference. Thanks for noting this point. We wrote:

*"Similarly to other studies (for example, Rohm et al., 2019), we estimate the ZHD from the WRF surface pressure because no pressure observations were available in correspondence of the GNSS-ZTD receivers."*

• Case study analysis: The case study results are examined on the basis of maps comparison. Please provide a statistical evaluation of the results (as in the whole period analysis).

Yes, we agree with the reviewer. We put the precipitation score for the case study in the supplemental material of the paper. Instead of the performance diagrams, we showed the FBIAS, ETS, POD and FAR for different precipitation threshold (1mm/3h and from 2 to 60 mm/3h every 2mm/3h). We considered the 0-3h phase only.

• Results and discussion: Please enrich this section in terms of interpreting the results and placing them in in the context of the related literature.

Ok. We have enriched the discussion putting it better into the context.

Minor points

Overall, the paper is well written and structured, but some points can be improved. These include the frequent use of separate lines instead of longer paragraphs. For instance, the Abstract should be a single paragraph. Further,

• Abstract: Lines 23-25 (about the results) should be placed before Lines 21-22 (about the results). Further, it seems that there is repetition in Lines 21-22 and 28-30. Please be more specific

concerning "showed an improvement of the precipitation forecast in different ways". Please revise "model 4.1.3" to "model, version 4.1.3,".

We revised the abstract taking into considerations the comments above. Now the difference between water vapour analysis and precipitation analysis should be more defined, and the logical sequence of the sentences should be smoother.

• Introduction: The first paragraph lacks a conceptual connection to the content provided in the next paragraphs. This is also true for the sentence in Lines 68-69 (please refer to the countries) in relation to the previous paragraph. Please make clear that the studies of Lagasio et al. (2019) and Mascitelli et al. (2019, 2021) were performed over Italy (Lines 76-81). Please provide abbreviations for the terms 3DVar, 4DVar etc.

We added a sentence to link better the first and second paragraphs, that were also joined. We added the terms abbreviations. Finally we put better in the context the references and countries of lines 68-69. Done for Lagasio and Mascitelli papers.

• Lines 112-113: The "background simulations" usually refer to those for deriving the model background errors for DA. Thus, I suggest renaming the experiments without DA to "control simulations".

Ok, we implemented this naming. Note that we used CTRL and GNSS replacing BCKG and GPS labels to indicate control and GNSS data assimilation experiments.

• Please provide a map in Appendix A showing the locations of the rain gauges used for evaluating the model results.

We added the following figure in Appendix A:

[Figure]

Figure: Elevations of the raingauges.

• Please specify the reason of examining the innovations during the case study analysis (Lines 248-262). To my understanding this is done to highlight that the ZTD DA assimilation leads to actual modeled differences related to PWV that are not random.

The reason for showing the innovation for the case study is that the difference between the precipitation with or without data assimilation can be well interpreted considering these innovations. A more detailed explanation of the point has been given in the paper. We wrote:

"*We start examining the innovations, i.e. the analysis minus first guess fields, at 18 UTC on 15 October (Figure 5a), which is the last analysis before the 3h forecast considered in this section and has an important role on the 3h rainfall forecast. Indeed, as shown below, the precipitation between 18 and 21 UTC has several correspondences with the innovations at 18 UTC.*"

In addition, the complex pattern of the innovations reveals, although indirectly, the huge number of GNSS receivers used for the analysis. However, this is already written into the paper and no addition was done in this direction.

- Figure 5: Please indicate the axis of the cross-section.

We used the following figure and we adjusted the figure caption accordingly:

[Figure]

References

Lagasio, M., Parodi, A., Pulvirenti, L., Meroni, A.N., Boni, G., Pierdicca, N., Marzano, F.S., Luini, L., Venuti, G., Realini, E.,Gatti, A., Tagliaferro, G., Barindelli, S., Monti Guarnieri, A., Goga, K., Terzo, O., Rucci, A., Passera, E., Kranzlmueller, D., Rommen, B.: A Synergistic Use of a High-Resolution Numerical Weather Prediction Model and High-Resolution Earth Observation Products to Improve Precipitation Forecast. Remote Sens.-Basel, 11, 2387. https://doi.org/10.3390/rs11202387, 2019.

Mascitelli, A., Federico, S., Fortunato, M., Avolio, E., Torcasio, R. C., Realini, E., Mazzoni, A., Transerici, C., Crespi, M., Dietrich, S.: Data assimilation of GNSS-ZTD into the RAMS model through 3D-Var: preliminary results at the regional scale. Meas. Sci. Technol. 30, 055801 (14pp). https://doi.org/10.1088/1361-6501/ab0b87, 2019.

Mascitelli, A., Federico, S., Torcasio, R. C., Dietrich, S.: Assimilation of GPS Zenith Total Delay estimates in RAMS NWP model: Impact studies over central Italy. Adv. Space Res., https://doi.org/10.1016/j.asr.2020.08.031, 68, 12, pp 4783-4793, 2021.

**References (will be added to the paper)**

Krietemeyer, A.; Ten Veldhuis, M.-c.; Van der Marel, H.; Realini, E.; Van de Giesen, N. Potential of Cost-Efficient Single Frequency GNSS Receivers for Water Vapor Monitoring. *Remote Sens.* 2018, *10*, 1493. https://doi.org/10.3390/rs10091493

Tagliaferro G.: On the Development of a General Undifferenced Uncombined Adjustment for GNSS, PhD Dissertation 2021, Politecnico di Milano, Department of Civil and Environmental Engineering.

Answers to reviewer n°2

The paper studies the impact of GNSS data assimilation over Italy with a focus on a 6-hour forecast based on precipitable water vapor and precipitation. It is evident from the study that GNSS DA improves the underestimation of water vapor by WRF.

We acknowledge the reviewer for reviewing the paper and for the useful comments. In the following our answers are in red, while the parts of the paper are referred in blue.

I mostly agree with the points mentioned by **Referee #1**, however, I think that reframing the aim of the paper and adding some additional analysis that could bring some novelty to the paper. From the reply comments of the author to the reviewer, I see the author has already started to prepare results in the direction of data thinning experiments which is good.

In the revised version of the paper, we discuss the results of the data thinning experiment and we reframe the sentence about the novelty of the paper. In addition, we add the results of an experiment clarifying the role of the bias removal.

 My major concern to add is that the data assimilation experiments need some more elaboration:

- regarding the observations assimilated in the BCKG experiment.

We rephrased the sentences in sections 2.1 on this point, better highlighting than BCKG doesn't use data assimilation. We wrote:

"We consider two kinds of simulations: control simulations, without GNSS-ZTD data assimilation, hereafter also CTRL, and simulations assimilating GNSS-ZTD, hereafter also GNSS. The European Centre for Medium range Weather Forecast (ECMWF) Integrated Forecast System (IFS) 3-hourly operational analysis/forecast cycle at 0.25° starting at 12 UTC on the day before the actual day to forecast is used for initial and boundary conditions, to simulate a real forecasting context. The temporal scheme used for the simulations uses a Very Short-term Forecast (VSF) approach, with a 6h update (Figure 2).

[Figure]

Figure 2: Rapid Update Cycle at 6h. Red dots denote analysis times.

In this scheme, for each day, we run four simulations starting from a cold start.  Each simulation lasts 12 hours. The first 6h of each run are used for the model spin-up and for data assimilation in GNSS simulations, while the last 6h are used as forecast. Therefore, 4 runs are necessary to cover a whole day. For GNSS simulations, in the assimilation phase, we considered an analysis every 1

hour (red points of Figure 2), starting from the beginning of the simulation and reaching the 6th hour, so a total of 7 analyses are performed for each run. For CTRL simulations we use only initial and boundary conditions from ECMWF-IFS and no other data are assimilated."

- regarding the GNSS data used for assimilation and if bias correction was performed before the assimilation of GNSS data.

Thank you for this comment, which gives the opportunity to clarify the point. We applied the bias corrections before data assimilation using the following procedure. To consider this point we wrote in Section 2.1:

"The GNSS-ZTD observations are considered to have unbiased errors compared to the WRF model. To achieve this goal, a statistical bias correction was applied to the ZTD data for the whole period. First the raw GNSS data are assimilated in the 3DVAR to calculate the corrections that come from the background. The difference between the observation and the background is saved for each receiver and for each time giving the quantity $(O-B)_{k,t}$, where $k$ is the receiver index and $t$ is the time. The quantity $(O-B)_{k,t}$ takes into account for the difference between the model orography and receiver height that, in our case, is never larger than 300 m. For each receiver we compute the background bias by averaging $(O-B)_{k,t}$ over all times:

$$\overline{(O-B)_k} = \sum_{t=1}^{N} \frac{(O-B)_{k,t}}{N}$$

Where $N$ is the total number of times (i.e. observations) available for each GNSS receiver. Then we use the corrected observation $O'_{k,t}$ in the 3D-Var:

$$O'_{k,t} = O_{k,t} - \overline{(O-B)_k}$$

This method is like that applied in many papers including some cited in the Introduction."

My minor concern to add is about the tense usage in the text throughout the journal should be uniform. Also, the paragraph construction should be more refined regarding the main point to express in a context.

We have reviewed the tense usage.

---

## Author Response (AR2)

We acknowledge the reviewer for the work on the paper.

Reviewer comment:
I would like to thank the authors for addressing all my initial comments. According to their responses, the conceptualization and novelty of the study (my major concern) are better justified. However, this is not yet clearly presented in the manuscript (Lines 111-115). I suggest enriching this part, in order to better highlight the differences with the previous studies in Italy and show that the present study builds upon previous literature and goes further, rather than going in "a similar direction". The operational context under which the investigation is conducted should be also highlighted. In this direction, please also include in the manuscript the fact that the applied model configuration is similar to the one used by CNR-ISAC for providing weather forecasts to the Department of Civil Protection in Italy.

Answer:
We changed the lines 111-115:
"This paper goes in a similar direction in the sense that it uses the GNSS-ZTD data assimilation to improve the precipitation and water vapor forecast over Italy. It uses a period of one month (October 2019) and the data of 388 GNSS receivers widespread over the country, giving a robust assessment on the impact that GNSS-ZTD data assimilation can have on the forecast at the local scale. In addition, for the first time over Italy, the sensitivity of the results to the number of GNSS receivers used (data thinning) and to the bias correction are shown for a subperiod of 16 days (5-10 and 14-23 October)."

to:

This paper enriches the numerical experiments made over Italy to improve the precipitation and water vapor forecast through GNSS-ZTD data assimilation because it refers to a different period compared to previous studies (October 2019) and uses data of 388 GNSS receivers widespread over the country for the whole period, giving a robust assessment of the impact that GNSS-ZTD data assimilation can have on the forecast at the local scale. In addition, it considers two issues that are important in the operational context: the optimal spacing of GNSS receivers for data assimilation, and the bias removal. The first experiment estimates the observation error decorrelation length scale and applies it to optimize the GNSS-ZTD data assimilation by data thinning, while the second experiment quantifies the impact of the bias removal on the forecast performance, because the bias cannot be completely removed in an operational context.
It is also noticed that a similar configuration of the WRF model used in this paper is already operational at the CNR-ISAC in the framework of the agreement between the Department of Civil Protection (DPC) and CNR-ISAC to improve the NWP forecast at different time ranges. So, the results of this paper are of practical importance as GNSS-ZTD data could also be assimilated in the near future in the operational run.